# Expanding automated multiconformer ligand modeling to macrocycles and fragments

Jessica Flowers[1], Nathaniel Echols[1], Galen J Correy[1], Priyadarshini Jaishankar[2], Takaya Togo[2], Adam R Renslo[2], Henry van den Bedem[1,3], James S Fraser[1]*, Stephanie A Wankowicz[1]*†

[1]Department of Bioengineering and Therapeutic Sciences, University of California, San Francisco, San Francisco, United States; [2]Department of Pharmaceutical Chemistry, University of California, San Francisco, San Francisco, United States; [3]Atomwise Inc, San Francisco, United States

*For correspondence:
jfraser@fraserlab.com (JSF);
stephanie@wankowiczlab.com
(SAW)

Present address: †Department of Molecular Physiology and Biophysics, Vanderbilt University, Nashville, United States

## eLife Assessment

The work presents a **valuable** extension of qFit-ligand, a computational method for modeling conformational heterogeneity of ligands in X-ray crystallography and cryo-EM density maps. The authors provide **solid** evidence of improved capabilities through careful validation against the previous version, particularly in expanding ligand sampling within conformational space. Such improvements suggest practical utility for challenging applications, including macrocyclic compound modeling and crystallographic drug fragment screening.

**Abstract** Small molecule ligands exhibit a diverse range of conformations in solution. Upon binding to a target protein, this conformational diversity is reduced. However, ligands can retain some degree of conformational flexibility even when bound to a receptor. In the Protein Data Bank, a small number of ligands have been modeled with distinct alternative conformations that are supported by macromolecular X-ray crystallography density maps. However, the vast majority of structural models are fit to a single-ligand conformation, potentially ignoring the underlying conformational heterogeneity present in the sample. We previously developed qFit-ligand to sample diverse ligand conformations and to select a parsimonious ensemble consistent with the density. While this approach indicated that many ligands populate alternative conformations, limitations in our sampling procedures often resulted in non-physical conformations and could not model complex ligands like macrocycles. Here, we introduce several improvements to qFit-ligand, including integrating RDKit for stochastic conformational sampling. This new sampling method greatly enriches low-energy conformations of small molecules and macrocycles. We further extended qFit-ligand to identify alternative conformations in PanDDA-modified density maps from high-throughput X-ray fragment screening experiments, as well as single-particle cryo-electron microscopy density maps. The new version of qFit-ligand improves fit to electron density and reduces torsional strain relative to deposited single-conformer models and our prior version of qFit-ligand. These advances enhance the analysis of residual conformational heterogeneity present in ligand-bound structures, which can provide important insights for the rational design of therapeutic agents.

## Introduction

Protein–ligand interactions are fundamental to many biological processes, involving both natural metabolites that regulate proteins and drugs developed to activate or inhibit proteins for therapeutic purposes. Prior to binding, both the ligand and protein receptor can sample a wide number of conformations. Upon binding, it is typically assumed that both ligand and protein will lose access to nearly all of their conformational states (*Chang et al., 2007*). This assumption leads to the common practice in X-ray crystallography and single-particle cryo-electron microscopy (cryo-EM) of modeling the ligand as adopting a single, fixed conformation within the binding site, with little to no consideration of potential heterogeneity other than refined B-factors.

Both macromolecular X-ray crystallography and cryo-EM generate averaged datasets by compiling scattering information from >10,000s of system copies, including macromolecules, solvents, ions, and small molecules. The resulting data from which structural models are built encompass significant conformational and compositional heterogeneity (*Wankowicz and Fraser, 2024a*). Conformational heterogeneity, when the same substance is in multiple conformations, includes subtle, sub-Ångstrom changes that are difficult to model by eye, yet these shifts are crucial for accurate biological interpretation (*Wankowicz et al., 2022*). Compositional heterogeneity refers to variation in the molecular contents of a sample, such as a ligand or macromolecular subunit only bound in a portion of the complexes captured. Ligand modeling, even as a single conformer, is challenging due to compositional heterogeneity, interference from water molecules, and system-wide conformational heterogeneity, all of which lead to ambiguity in electron density map interpretation (*Nicholls, 2017*). This challenge in manual modeling is a major reason why structural variability is often underrepresented in deposited models. However, in the Protein Data Bank (PDB), a small number of ligands are modeled as multiple conformers, representing their conformational heterogeneity (*van Zundert et al., 2018*; *Liebeschuetz, 2021*). These structures likely represent just a small fraction of ligands with experimental evidence that could support modeling multiple conformations, as has been shown in proteins (*Wankowicz et al., 2022*; *Smith et al., 1986*; *Wankowicz et al., 2024c*). When handled correctly, modeling ligands in multiple conformations can reveal critical information about biological function (*Díaz et al., 2024*) and guide small molecule design (*Zhao et al., 2023*; *Mehlman et al., 2024*).

To help assist in modeling conformational heterogeneity, we have developed qFit, which can automatically build multiconformer models (*Wankowicz et al., 2024c*; *van den Bedem et al., 2009*; *Keedy et al., 2015*; *Riley et al., 2021*). The underlying concept of qFit is to enumerate a large number of conformations according to a sampling procedure and then to use mixed integer quadratic programming (MIQP) to optimize the selection of a parsimonious set of conformers, along with their corresponding occupancies (*van den Bedem et al., 2009*). This approach improves the fit to experimental data and agreement with geometric priors for proteins (*Wankowicz et al., 2024c*). We previously extended qFit to qFit-ligand to identify and model alternative conformations of ligands to experimental data (*van Zundert et al., 2018*).

The prior version of qFit-ligand used iterative sampling over each torsional degree of freedom (*van Zundert et al., 2018*). This approach overlooked correlated motions and over-explored conformations that were energetically unfavorable. Here, we present a redeveloped sampling algorithm powered by the RDKit implementation of the Experimental-Torsion Knowledge Distance Geometry (ETKDG) conformer generator, which is a stochastic search method that combines distance geometry and knowledge derived from experimental structures (*Wang et al., 2020*; *Riniker and Landrum, 2015*). We demonstrate that our improved qFit-ligand can automatically model multiple conformations of ligands where supported by electron density. The majority of qFit models improved real space correlation coefficients (RSCC), electron density support for individual atoms (EDIA), and ligand strain. We also extend qFit-ligand to accommodate emerging strategies in structure-based drug design, including macrocycles, fragment screening, and cryo-EM. While the cyclic nature of macrocycles makes modeling the flexibility by our prior approach incredibly troublesome, with improved sampling, we can now model this expanding class of small molecules (*Kamenik et al., 2018*). Second, X-ray-based fragment screening has exploded in popularity since our first release; however, these approaches rely on density map manipulations accounting for compositional heterogeneity (*Pearce et al., 2017*). With improved map handling, we can now model into these 'event' maps, identifying multiple conformations even for low molecular weight compounds. Lastly, recent advances in cryo-EM have enabled increasingly high-resolution reconstructions, which in turn allow for atom-level modeling

of conformational heterogeneity (*Cushing et al., 2024*). In response, we introduce cryo-EM map compatibility, making qFit-ligand a method for automated multiconformer ligand model building using cryo-EM data. Together, these advancements and the enhanced code base will enable more accurate identification and modeling of ligand conformational heterogeneity across a variety of ligands, leading to a better interpretation of protein–ligand interactions.

## Results

### Overview of the qFit-ligand algorithm

The qFit-ligand algorithm takes as input a crystal or cryo-EM structure of an initial protein–ligand complex with a single-conformer ligand in PDBx/mmCIF format, a density map or structure factors (encoded by a ccp4 formatted map or an MTZ), and a SMILES string for the ligand. The SMILES string is used for bond order assignment internally (Methods). The algorithm produces a multiconformer model of the ligand, embedded into the context of the rest of the unaltered structural model. This version of qFit-ligand leverages advances to the code base that have improved the stability of the code for protein modeling applications (*Wankowicz et al., 2024c*) and uses the Chem.rdDistGeom module of RDKit, which implements ETKDG, for conformational sampling (*Wang et al., 2020*; *Riniker and Landrum, 2015*) (see Conformer Generation).

To ensure compatibility with the surrounding protein, the ensemble is generated under constraints defined by the geometry of the binding site (see Biasing conformer generation), generating 5000–7000 ligand conformations depending on the size of the ligand. We then use quadratic programming (QP) and MIQP optimization algorithms to determine the best fit of the coordinate and occupancy of conformers to the experimental map. For X-ray data, we restrict the algorithm to output a maximum of three conformations, whereas cryo-EM is restricted to outputting a maximum of two conformations. Examples can be found in the qFit Github repository (https://github.com/ExcitedStates/qfit-3.0 copy archived at *Riley et al., 2025*) under version 2025.1 and is packaged as part of SBGrid (*Morin et al., 2013*).

### Conformer generation

For an input molecule (*Figure 1*), the RDKit Chem.rdDistGeom.EmbedMultipleConfs function generates a distance bounds matrix containing the minimum and maximum allowable distances between every pair of atoms for an input molecule (*Figure 1—figure supplement 1*; *Blaney and Scott Dixon, 2007*). The algorithm then explores the conformational space by stochastically generating distances within the defined distance bounds, generating diverse and chemically plausible conformers across torsional angles (*Figure 1A–E*). For example, within a torsion angle formed by four atoms, the minimum distance between atoms 1 and 4 corresponds to the syn conformation, and the maximum distance corresponds to the anti conformation. Conformations are generated with torsional angles between the maximum and minimum, while obeying other constraints, ensuring exploration of the molecule's conformational space within realistic and chemically meaningful limits.

The sampled distances are converted into three-dimensional coordinates through an embedding procedure. Next, torsional angles are refined using potentials derived from experimental distributions observed in the Cambridge Structural Database (CSD) (*Allen, 2002*; *Groom and Allen, 2014*) (Methods). Following torsional minimization, we apply the optionally available force field minimization step, using the MMFF94 force field (*Tosco et al., 2014*) to eliminate steric clashes and reduce molecular strain (*Wang et al., 2020*; *Riniker and Landrum, 2015*). All these steps help to ensure that only conformers with low torsional strain are allowed to be selected for final fitting.

### Biasing conformer generation

To guide the conformation generation from the Chem.rdDistGeom based on the ligand type and protein pocket, we developed a suite of specialized sampling functions to bias the conformational search toward structures more likely to fit well into the receptor's binding site. For a given molecule, up to six of these modified sampling functions are used to refine the conformational search. All steps are initialized with the input ligand model and are run in parallel.

First, in all cases, we perform an *unconstrained search function* (*Figure 1A*), a *fixed terminal atoms search function* (*Figure 1B*), and a *blob search function* (*Figure 1C*). The *unconstrained search function*

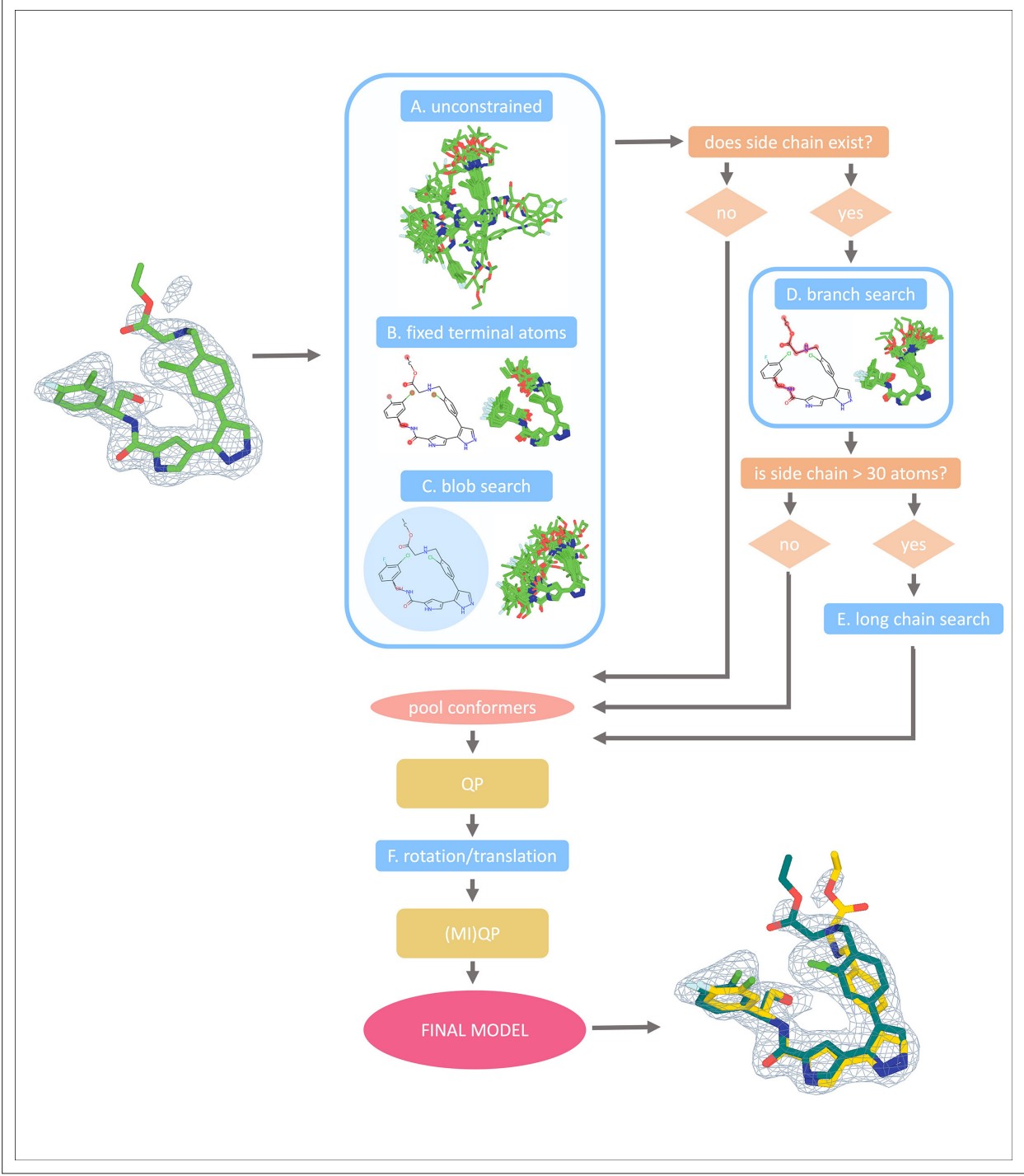

**Figure 1.** qFit-ligand algorithm workflow. All ligands undergo three preliminary searches: unconstrained, fixed terminal atoms, and blob search, allowing varying degrees of freedom (**A–C**). If the ligand has short or long side chains, the algorithm progresses to more specialized searches: branch search for ligands with side chains of at least four atoms (**D**), and long chain search for those exceeding 30 atoms (**E**). The algorithm then determines the best fit of generated conformers to electron density through quadratic programming, followed by additional sampling with rotations and translations (**F**). The remaining conformers then undergo quadratic and mixed-integer quadratic programming to ensure that only the most well-supported conformers are included in the final model.

The online version of this article includes the following figure supplement(s) for figure 1:

**Figure supplement 1.** RDKit determines a distance bounds matrix for a molecule by establishing upper and lower bounds for interatomic distances.

**Figure supplement 2.** Correlation between the number of atoms in the input ligand and total qFit-ligand runtime.

generates conformers only constrained from the default RDKit parameters as described above. The *fixed terminal atoms search function* places hard constraints on the distance between the terminal atoms, allowing the atoms in between to randomly sample distances within their respective upper and lower bounds. This preserves the overall shape of the ligand while still allowing for internal movement. Finally, the *blob search function* confines generated conformers within a spherical volume, determined by the maximum Euclidean distance from the geometric center of the input ligand to its outermost atoms.

For ligands with side chains of at least four atoms, we also implement a *branching search function* (*Figure 1D*). Here, atoms not included in the side chain (core atoms), are fixed to the coordinates of the input ligand model. This method allows the sampling of side chain conformations while maintaining the relative positioning of the core atoms. When these chains exceed 30 atoms, we apply a *long chain search function* (*Figure 1E*). This approach does the opposite of the *branching search function* by fixing the atoms in the long side chains in place while allowing the core atoms to explore various conformations. This ensures the generation of relevant conformations of the core atoms without excessive variability in the side chains, which is crucial for ligands with a high degree of freedom.

Additionally, an optional flag turns on the *180° flip sampling function*. This function takes the input modeled ligand conformer and rotates it 180° around the three principal axes (*x*, *y*, and *z*), effectively generating three new conformations that are flipped relative to the original structure. After each 180° flip, the function applies rotations to each of these three conformers within a range of ±10° in 2° increments. This option is turned off by default and is only recommended for supervised cases where a user suspects the ligand in their crystal may adopt this specific type of conformational disorder.

By default, each run of qFit-ligand generates 5000 conformers if the input ligand has fewer than 25 heavy atoms and 7000 otherwise, evenly distributed across the specialized search strategies. Users can optionally customize this number using the command line flag '-nc'. After all conformers are generated, we identify pairs of redundant conformers, defined as those with a root mean square deviation (RMSD) less than 0.2 Å, randomly choosing one to remove.

To select the set of conformers that best explains the observed density, qFit-ligand employs a QP optimization algorithm. For each sampled conformer, we generate a calculated density map based on the ligand's atomic coordinates, element types, B-factors, and the map resolution. Each conformer is assigned a weight (occupancy) that collectively optimizes the real space residual of the observed density versus the weighted sum of all the calculated densities. The algorithm has two constraints, first that all weights are non-negative and that the sum of all weights lies between 0–1. QP usually outputs 1–80 conformations (Methods). We then further sample these remaining conformers by applying rotational and translational perturbations (*Figure 1F*). New conformations are created by rotating by 15° in 5° increments and translating by 0.3 (Å) along the *x*, *y*, and *z* axes. Conformers are then selected through an additional round of QP. The final conformations are then selected using MIQP, where the optimization problem is the same (optimizing real space residuals of observed versus weighted sum of all calculated densities), but with additional linear constraints to limit the final multiconformer model to a maximum of three (or two for cryo-EM) conformers. The output is then one to three ligand conformations with relative occupancies that collectively best explain the observed density (Methods).

## Refinement of qFit-ligand models

qFit-ligand builds a parsimonious multiconformer ligand model and outputs both an independent ligand structure and the protein–ligand complex embedded in the rest of the system (containing solvent, other heteroatoms, etc). After running qFit-ligand, we refine this complex using phenix.refine (*Afonine et al., 2012*) or phenix.real_space_refine for cryo-EM structures (*Afonine et al., 2018*). The resulting final, refined model is used for all subsequent comparisons throughout the rest of the paper.

## qFit-ligand runtime

qFit-ligand operates on up to five CPU cores, demonstrating efficient performance on a standard laptop, if all five cores are engaged, with typical runtimes for most ligands (70.6%) being less than 10 min (mean: 8.6 min, median: 6.1 min, range: 1.9–44.9 min). qFit-ligand is not parallelized by default, but an optional command line flag '-p' is available to set the number of cores used during conformer generation. Analysis across a large dataset of structures reveals a strong correlation between the

size of the input ligand and the runtime (Pearson correlation coefficient of 0.75), with larger ligands resulting in longer processing times (*Figure 1—figure supplement 2*).

## Detection of experimental true positive multiconformer ligands

To develop the new qFit-ligand algorithm, we collected a set of true positive multiconformer ligand models from the PDB. We identified 2,199 PDB files containing ligands with multiple conformations, more than 10 heavy atoms, and resolutions better than 2.0 Å. We removed structures that had alternative conformers in common crystallographic additives (*n* = 453), as well as structures with the same protein and ligand pair (*n* = 212). This further pruned our collection to 1,534 structures, with resolutions ranging from 0.73 to 1.99 Å. We randomly sampled 150 structures and after a manual inspection, removed 15 where the deposited conformations did not visually resolve well into the density, leaving us with 135 structures as a development set for improving qFit-ligand (*Figure 2—figure supplement 1*, *Supplementary file 1, table 1*).

To simulate a realistic scenario where the multiple conformations of a ligand are initially unknown, we retained only the 'A' conformations (all structures had 2 conformations), setting its occupancy to 1.00. Occupancy of the 'A' conformer was higher than the 'B' conformer in 82.2% of structures (*n* = 111/135). These single-conformer ligand structures were refined using phenix.refine (*Afonine et al., 2012*) (Methods). We refer to these altered structures as our 'modified true positives', which we use as input to qFit-ligand, and subsequent refinement using Phenix (Methods) (*Afonine et al., 2012*). For the map input, we calculated a $2mF_o − DF_C$ composite omit map using the modified true positive model and the deposited structure factors. A composite omit map is a crystallographic density map that reduces model bias by omitting small regions of the model, calculating the density for each omitted segment, and then combining the results into a final map (*Terwilliger et al., 2008*).

To evaluate the impact of qFit-ligand algorithmic improvements, we compared the modified true positive dataset to the output of qFit-ligand (qFit-ligand dataset), evaluating three primary metrics: RSCC, EDIAm, and ligand torsion strain (Methods). RSCC evaluates how well the model fits into the electron density, with values exceeding 0.80 indicating a satisfactory agreement between the model and experimental data (*Smart et al., 2018*; *Shao et al., 2022*). EDIAm assesses the local agreement between atomic positions and the electron density map, providing a more sensitive per-atom measure of model quality, where higher values indicate stronger support from the experimental data (*Meyder et al., 2017*). Torsion strain measures the physical viability of predicted conformations, where lower strain values suggest more stable and naturally occurring conformations. To carry out these strain calculations, we use the software *TLDR: Strain* (*Gu et al., 2021*), which calculates ligand strain by comparing the torsional angle populations of a ligand to those in the CSD, quickly assessing strain energy without detailed quantum or molecular mechanical calculations. This is a different strain calculation than what is used internally in RDKit, ensuring that this is a somewhat independent metric. We note that there is currently no consensus in the field regarding what constitutes a significant improvement in RSCC, EDIAm, or strain, but we believe that any marginal improvements likely reflect a model that more accurately reflects the underlying experimental data.

qFit-ligand modeled an alternative conformation in 72.5% (*n* = 98) of structures. Compared with the modified true positive models, 83.7% (*n* = 113) of qFit-ligand models have a better RSCC, and 77.0% (*n* = 104) structures saw an improvement in EDIAm, representing an improved fit to experimental data in the vast majority of structures. Further, the majority of structures (61.5%, *n* = 83) exhibited reduced torsional strain with qFit-ligand, with a mean difference of –0.2 kcal/mol (*Figure 2A, B*, *Figure 2—figure supplement 2*). This suggests that over half of the qFit-ligand models were more energetically favorable compared to the modified true positive models, however, the majority of these improvements were of relatively low magnitude. The increased strain of the modified true positives may be due to the removal of correctly modeled alternate conformations observed in the deposited structures, followed by re-refinement of an incomplete single-conformer model using Phenix. Thus, the reduced strain observed in our qFit-ligand models relative to the modified true positives is not unexpected. Overall, 48.9% (*n* = 66) of ligands had both improved RSCC and reduced torsional strain, demonstrating that we frequently improved the fit between experimental data, while also maintaining or improving the strain.

To identify places for algorithmic improvement, we examined the five qFit-ligand structures for which there was the greatest degradation in strain compared to the input model. In most cases,

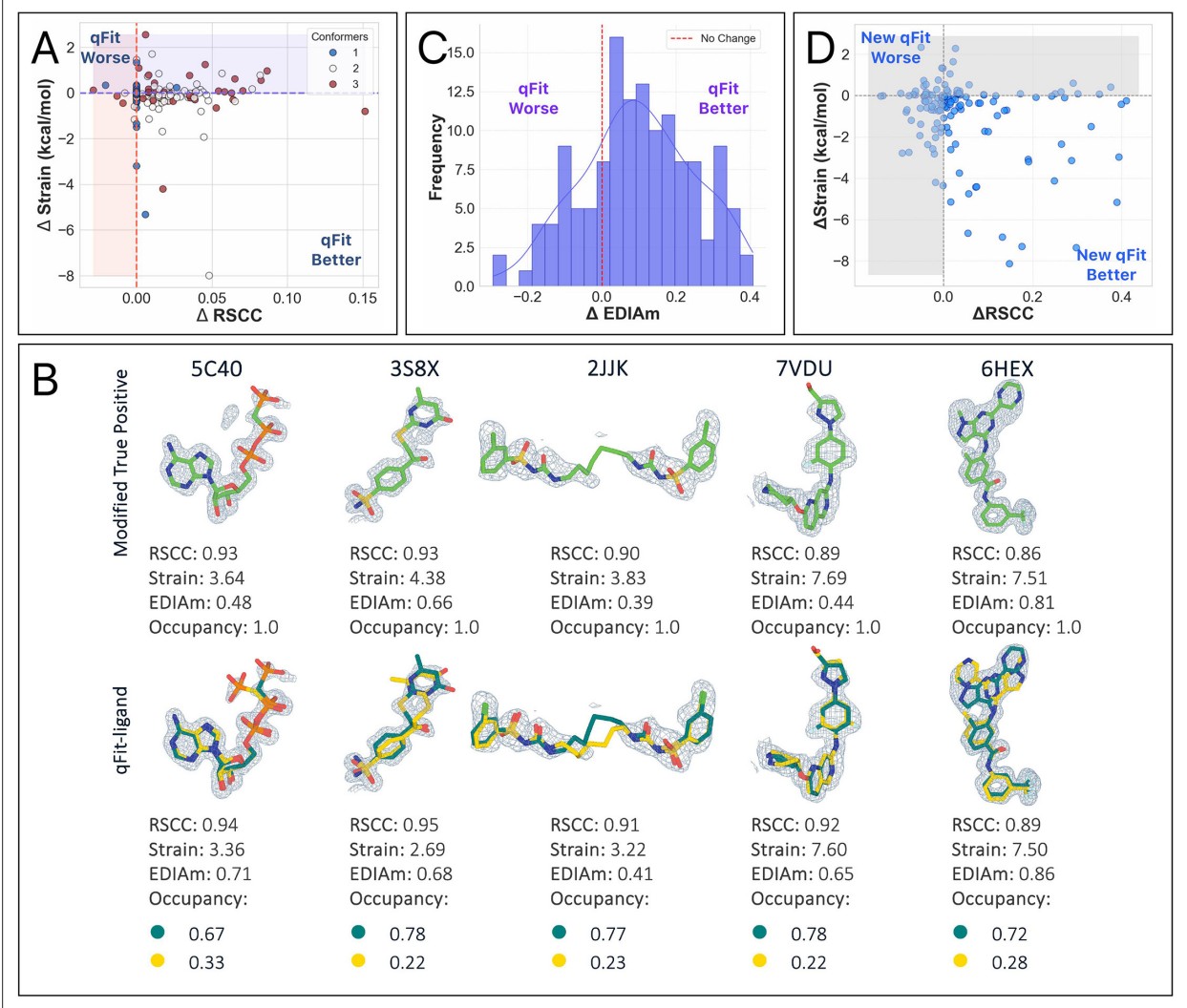

**Figure 2.** Analysis of ligand conformations generated by qFit-ligand. (**A**) Differences in real space correlation coefficients (RSCC) (*x*-axis) and torsion strain (*y*-axis) between qFit-ligand predicted structures and modified true positives. The lower right quadrant shows structures for which we improve both RSCC and strain. (**B**) Gallery of examples for which the new qFit-ligand models have improved RSCC, strain, and EDIAm compared to the modified true positives. The composite omit density map is contoured at 1σ for every structure. (**C**) Differences in EDIAm between qFit-ligand models and modified true positives. Positive delta values indicate structures where the qFit-ligand model is a better fit to the experimental density. (**D**) Differences in RSCC and torsion strain between the new qFit-ligand and the prior qFit-ligand. The lower right quadrant shows structures for which we improve both RSCC and strain.

The online version of this article includes the following figure supplement(s) for figure 2:

**Figure supplement 1.** Construction of the development true positive dataset and the unbiased true positive dataset.

**Figure supplement 2.** Original (unmodified) multiconformer true positives compared to qFit-ligand conformers.

**Figure supplement 3.** Comparison of torsion strain between qFit-ligand models before and after refinement, as well as the deposited structures.

**Figure supplement 4.** Performance comparison of new and prior qFit-ligand algorithms.

**Figure supplement 5.** Modified true positive dataset comparison of new versus prior qFit-ligand outlier cases.

the unrefined qFit-ligand model displayed strain levels that were much closer to the modified true positive, but strain increased after refinement with Phenix (*Figure 2—figure supplement 3*). While refinement improves the correlation between the model and the electron density map, it may inadvertently increase strain without careful calibration of geometry weights and restraint files. This should be carefully examined by the modeler.

To assess improvements over the prior version of qFit-ligand, we examined how the prior version performed on the modified true positive dataset. Compared to the prior version, we found that the new qFit-ligand achieved higher RSCC values in 57.8% ($n$ = 78) of the structures (*Figure 2—figure supplement 4A*), lower strain in 68.9% ($n$ = 93) (*Figure 2C*, *Figure 2—figure supplement 4B*), and higher EDIAm in 85.9% ($n$ = 116) (*Figure 2—figure supplement 4C, D*). We closely examined outlier cases where the new qFit-ligand most dramatically outperformed its predecessor. Among the 10 structures with the largest strain reduction and concurrent increase in RSCC and EDIAm, 6 of the deposited true positive models exhibited branching disorder, where a side chain in the ligand adopts an alternate conformation. In these models, the new qFit-ligand decreased strain by up to 8.1 kcal/mol, increased RSCC by up to 0.4, and increased EDIAm by up to 0.6. These examples highlight an improvement in our modeling of non-localized conformational disorder, where the structural heterogeneity affects large portions or the entirety of the ligand, often involving shifts in all atomic coordinates or branching side chains (*Figure 2—figure supplement 5*).

Interestingly, among the structures where the prior algorithm produced a model with a higher RSCC ($n$ = 56), 67.9% ($n$ = 38/56) were found to be higher in strain compared to the models created by the new qFit-ligand. This suggests that while the prior algorithm sometimes provided a better fit to the density, it often did so by compromising on structural or geometric integrity of the ligand. Moreover, of the structures where the prior qFit-ligand produced a model with a better RSCC ($n$ = 56), only 14.3% ($n$ = 8/56) had a new model RSCC lower than 0.80, indicating that the new qFit-ligand models were still generally well correlated to the experimental data. This demonstrates that the new qFit-ligand algorithm strikes a better balance between agreement with the density data and low-strain conformations. This directly addresses a major limitation in the prior version of qFit-ligand, which often produced conformers that fit the density but were physically or chemically unrealistic, as evidenced by their higher strain.

## Determining the operational bounds of qFit-ligand using synthetic data

To determine the lowest ligand occupancy qFit-ligand can accurately recognize and model across resolution ranges, we constructed a synthetic dataset comprised of four main ligand types. These include a ligand with a localized ring flip (3SC), a long linear ligand with non-localized displacement (3P3), one with localized disorder from a simple torsional shift (9BM), and a macrocycle with both branching and terminal end rotation heterogeneity (AR9) (*Figure 3—figure supplement 1*). For each ligand type, we designed an alternate conformation in COOT (*Emsley et al., 2010*) and created synthetic density data across a range of conformer occupancy ratios (0.50/0.50, 0.40/0.60, 0.30/0.70, 0.20/0.80, and 0.10/0.90) and map resolutions (0.8–2.5 Å, in 0.1 Å increments) (Methods). This resulted in 360 unique pairs of electron density maps and models, representing various combinations of conformer occupancy and resolution, which we refer to as the 'true' structures (*Figure 3—figure supplement 1*). We then inserted only the 'A' conformers into qFit-ligand to evaluate its ability to predict and approximate the 'B' conformer for each ligand type.

We directly compare the RSCC of the output qFit-ligand models with the true structures containing both conformers (*Figure 3A, B*). We observe a decrease in RSCC as resolution gets worse for all occupancy ratios. As map resolution approaches 2.0 Å, regardless of the occupancy split, there is a notable decline in qFit-ligand model RSCC. This suggests that qFit-ligand performs most effectively and consistently with map resolutions better than 2.0 Å.

While RSCC quantifies the overall map to model fit, our ultimate objective is the accurate recovery of alternate conformers. Therefore, we further utilized RMSD calculations to examine qFit-ligand's ability to recover the 'B' conformer present in the true model. We found that doing this successfully was correlated with the alternative conformer occupancy (*Figure 3C*). qFit-ligand models originating from a true model with an occupancy ratio of 0.50/0.50 and 0.60/0.40 exhibit comparable accuracy. Models with a 0.70/0.30 split begin to display marginally higher RMSD values, as well as an increase in inconsistency across map resolutions, though still remaining within acceptable limits. However, models at 0.80/0.20 exhibit greater variability across resolutions, with those at 0.90/0.10 showing even more pronounced inconsistencies. We show an example of the true versus qFit-ligand generated models for the 3SC ligand at a map resolution of 0.8 Å, with a true model conformer occupancy split of 0.50/0.50 and 0.20/0.80 (*Figure 3D*). These results suggest that the qFit-ligand occupancy detection limit is around 30%.

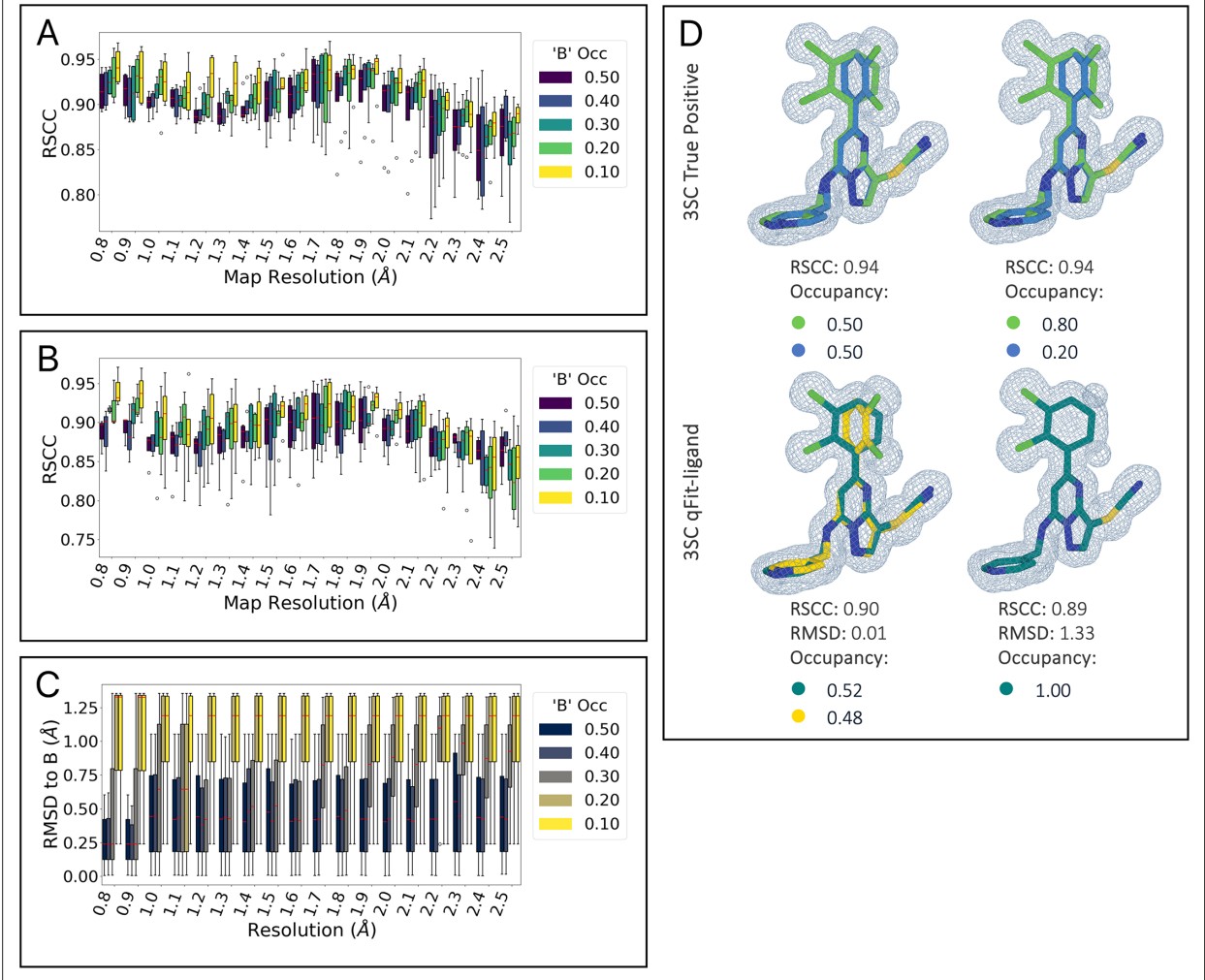

**Figure 3.** Resolution and occupancy limits of qFit-ligand. (**A**) Real space correlation coefficients (RSCC) of the synthetic true benchmark structures plotted against map resolution (in Ångstroms) for different conformer occupancy ratios, showing a decrease in RSCC with deteriorating map resolution. (**B**) RSCC of qFit-ligand generated multiconformer models, plotted against map resolution and grouped by conformer occupancy split. (**C**) Root mean square deviation (RMSD) between the closest qFit-ligand conformer and the true 'B' conformer. (D, left) True structure and qFit-ligand predicted structure of 3SC multiconformer ligand with a map resolution of 0.8 Å and conformer occupancy split of 0.50/0.50. (D, right) True structure and qFit-ligand predicted structure of 3SC multiconformer ligand with a map resolution of 0.8 Å and conformer occupancy split of 0.80/0.20.

The online version of this article includes the following figure supplement(s) for figure 3:

**Figure supplement 1.** The four ligand multiconformer models from which our synthetic dataset was built.

## qFit-ligand applied to unbiased dataset of experimental true positives

To determine how qFit-ligand performed on an independent dataset, we curated a new benchmark from the initial true positive collection of 1534 structures, excluding those used in the development set. Recognizing the impracticality of manually inspecting every structure and the detection limit we identified in the synthetic dataset, we applied additional filtering metrics to ensure data quality. Structures were required to have two deposited conformers with a root mean squared deviation (RMSD) of at least 0.2 Å, an average ligand B-factor of less than 80 $Å^2$, and conformer occupancies of at least 0.3. This process yielded a final set of 589 structures for analysis (*Figure 2—figure supplement 1*).

For all structures, we generated a modified true positive by deleting alternative conformers, setting occupancy to 1, and re-refining the model. We then followed the same outline as above, including pre-qFit refinement, qFit-ligand, and post-qFit refinement. The qFit-ligand models yielded 46.0% (*n* = 271) with a single conformer, 35.7% (*n* = 210) with two conformers, and 18.3% (*n* = 108) with three conformers (*Figure 4A*). Comparing qFit-ligand models to the modified true positives, 79.8% (*n* =

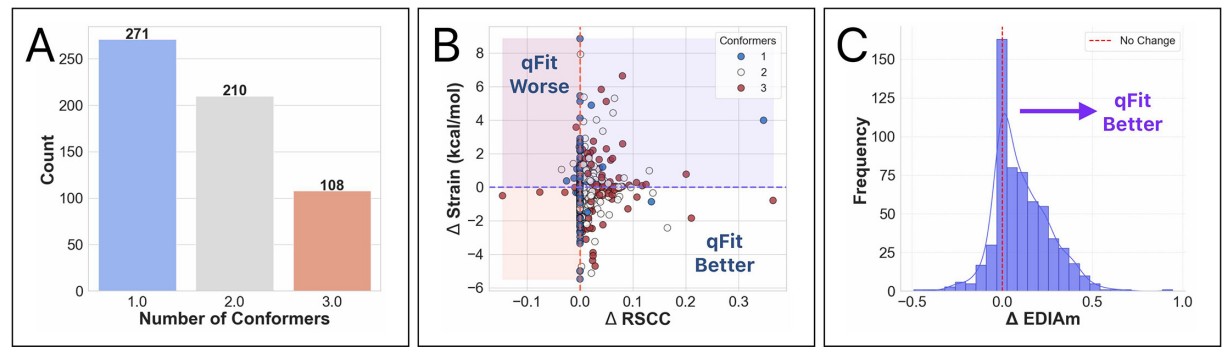

**Figure 4.** Analysis of ligand conformations generated by qFit-ligand on the unbiased modified true positive dataset. (**A**) Distribution of the number of conformers output by qFit-ligand. (**B**) Differences in real space correlation coefficients (RSCC) and torsion strain between the qFit-ligand models and the modified true positives. The lower right quadrant shows structures for which we improve both RSCC and strain. (**C**) Differences in EDIAm values between the qFit-ligand models and the modified true positives. Bars to the right of the vertical axis represent structures where the qFit-ligand model fits better to the electron density map.

470) showed an enhanced RSCC and 80.6% (n = 475) of the complexes had a higher EDIAm value (*Figure 4C*), reflecting a superior fit to the density map. qFit-ligand models had a reduced torsional strain in 55.3% (n = 326) of structures, though the overall strain difference was minimal in most cases (*Figure 4B*).

qFit-ligand shows particular strength in scenarios with strong evidence of unmodeled alternate conformations, often improving the fit to density, while sometimes improving the torsional strain. Interestingly, despite modeling a single conformer in nearly half of the structures, there is little to no evidence of qFit-ligand decreasing model-to-map fit. In fact, some of these single conformers show improved quality relative to the modified true positives (*Figure 4B*). These findings reinforce that qFit-ligand is not only capable of detecting alternate conformers when supported by the data, but also serves as a valuable alternative to manual ligand modeling even in single-conformer cases.

## Evaluating qFit-ligand on a set of structures known to be highly strained

High ligand strain is energetically unfavorable, and the associated energy penalty paid to adopt a distorted bound conformation reduces overall binding affinity (*Smola et al., 2021*; *Jain et al., 2023*). Because of this, it is generally accepted that drug-like molecules should adopt low-energy, minimally strained geometries. However, optimizing both fit to density and internal energetics simultaneously remains a challenge (*Liebeschuetz, 2021*). While our modified true positive datasets demonstrate that qFit-ligand can alleviate distortion by recovering alternate conformations, these cases are somewhat artificial; removing a valid altloc and re-refining can artificially inflate energetic penalties. To further validate our approach, we tested qFit-ligand on deposited structures with genuinely unfavorable conformations to gain a better understanding of whether our modeling algorithm impacts strain by discovering multiple low-energy conformations that satisfy the density as well, or better, than a single high-energy conformation.

To this end, we curated a dataset of deposited structures containing ligands with unusually high conformational energy by collecting all PDB entries with a resolution between 1.0 and 1.8 Å, an $R_{free}$ below 0.25, and a ligand molecular weight between 400 and 520 Da. We required the structures to not have a deposited alternate conformer. From an initial pool of 5452 structures, we followed the pre-qFit-ligand refinement protocol previously described, calculated ligand energies, and selected those exceeding 10 kcal/mol for our final dataset. This resulted in a collection of 191 structures that we used as input to qFit-ligand (*Supplementary file 2, table 2*). We note that there is no consensus in the field as to what constitutes high strain, but that 10 kcal/mol represents a conservative, if somewhat arbitrary cutoff (*Perola and Charifson, 2004*; *Sitzmann et al., 2012*; *Nicklaus et al., 1995*; *Borbulevych et al., 2018*; *Hao et al., 2007*; *Boström et al., 1998*; *Tong and Zhao, 2021*; *Rai et al., 2019*).

qFit-ligand modeled 75.4% (n = 144) of the structures with a single conformer, while 19.4% (n = 37) had two conformers, and 5.2% (n = 10) had three conformers (*Figure 5A*). Interestingly, even without

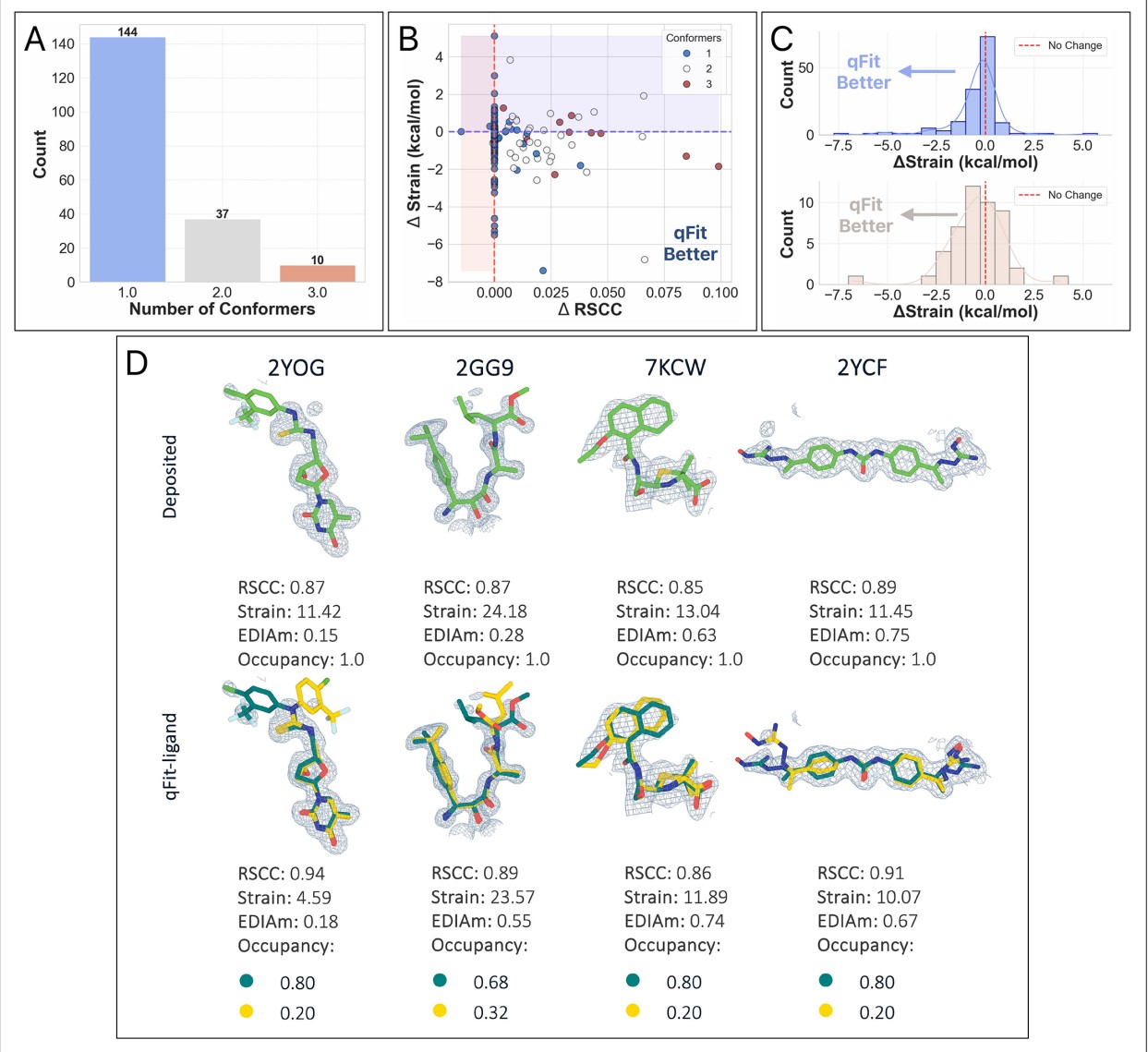

**Figure 5.** qFit-ligand improves fit of highly strained deposited molecules. (**A**) Distribution of the number of conformers modeled by qFit-ligand across 191 deposited structures with ligand torsional strain >10 kcal/mol. (**B**) Real space correlation coefficients (RSCC) and strain differences in the refined deposited models and the qFit-ligand predicted models. The lower right quadrant shows structures for which we improve both RSCC and strain. (C, top) Differences in torsion strain between the qFit-ligand models and the refined deposited models for structures where qFit-ligand predicted a single-conformer model. Negative delta values, all bars to the left of the vertical axis, represent structures for which the qFit-ligand model has a lower strain. (C, bottom) Differences in torsion strain between the qFit-ligand models and the refined deposited models for structures where qFit-ligand predicted a multiconformer model. Negative delta values, all bars to the left of the vertical axis, represent structures for which the qFit-ligand model has a lower strain. (**D**) Gallery of examples for which qFit-ligand successfully recovers well-fitting alternate conformers, and therefore reduces strain. The composite omit density map is contoured at 1σ for every structure.

The online version of this article includes the following figure supplement(s) for figure 5:

**Figure supplement 1.** Differences in EDIAm between the qFit-ligand models and the refined deposited models.

modeling an alternative conformer in the majority of structures, RSCC increased over the deposited model in 53.4% of structures ($n = 102$), EDIAm increased in 81.2% ($n = 155$), and strain decreased in 66.5% ($n = 127$) (***Figure 5B***, ***Figure 5—figure supplement 1***). Many of these strain improvements came from the pool of 144 single-conformer outputs, suggesting that qFit-ligand is able to sample from highly strained input models to result in new models that are out of a local minima (***Figure 5C, top***). In fact, the largest strain reductions in this dataset came from this pool of improved single-conformer models. We also identified several examples of qFit-ligand lowering strain through the

addition of a well-modeled alternate conformation (*Figure 5C, bottom; 5D*). In summary, our analysis reveals that qFit-ligand models adopt strain energies that are generally lowered from geometrically distorted deposited models, which supports using qFit-ligand to correct those distortions, even in the single-conformer case.

## qFit-ligand can automatically detect and model multiple conformations of macrocycles

While small molecules are great for inhibiting proteins with deep pockets, many proteins with pharmaceutical interests are classified as 'undruggable', due to their flat surfaces or involvement in protein–protein interactions. Macrocycles, cyclic molecules consisting of 12 or more atoms, have a great ability to interact with flat surfaces or shallow grooves (*Yudin, 2015*; *Driggers et al., 2008*; *Cummings and Sekharan, 2019*; *Russo et al., 2016*; *Garcia Jimenez et al., 2023*; *Vinogradov et al., 2019*). Due to their high degrees of freedom, the conformations of macrocycles are difficult to sample exhaustively and are likely to adopt a diverse ensemble in solution and even when bound to a receptor (*Appavoo et al., 2019*).

With our improved sampling strategy, we wanted to evaluate if we could accurately model multiple conformations of macrocycles. We utilized a dataset of 150 cyclic ligands with map resolutions ranging from 1.1 to 3.6 Å assembled during the development of XGen, an ensemble-based method for modeling macrocycles (*Jain et al., 2020*). Through an ensemble modeling strategy, XGen encodes several full-system copies that collectively satisfy the experimental data using restrained force field energy calculations. This procedure frequently reduced strain compared to input structures. In contrast, qFit-ligand represents conformational heterogeneity through a multiconformer approach, labeling discrete parsimonious conformations with alternative location indicators (altlocs). We wanted to determine if we could detect and explain the similar conformational heterogeneity as XGen using qFit-ligand and multiconformer models.

All the originally deposited macrocycle models contain only single-conformer ligands. As done above, we re-refined the deposited models before running qFit-ligand. Of these, 19.33% (*n* = 29) could not be refined against the deposited structure factors and were removed from the analysis. We then ran qFit-ligand as described in the methods section and re-refined output structures. Refinement is notoriously difficult for macrocycles due to difficulty creating restraint files. This can lead to altered chemical connectivity, effectively changing the ligand's composition, therefore, we conducted post-refinement ligand geometry validation checks to ensure that the chemical connectivity of the ligand remained unchanged, even if the conformation varied (Methods). We identified 19 cases of compromised ligand geometry (8 from pre-qFit and 11 from post-qFit refinement), which were subsequently excluded from this analysis. Additionally, strain calculation failed in 19.3% of cases (*n* = 29/150), producing N/A values, leaving 73 structures available for final analysis (*Supplementary file 3, table 3*, *Supplementary file 4, table 6*). Of note, the strain algorithm used was not developed for macrocycles, so this was not completely unexpected (*Gu et al., 2021*). The resolution range of the remaining structures was between 1.4 and 3.2 Å. The loss of structures reflects broader limitations in current bioinformatics and refinement protocols when handling complex macrocycle ligands.

Analysis of qFit-ligand models for these 73 macrocycles shows the following distribution of conformers per model: 39.7% (*n* = 29) having one conformation, 34.3% (*n* = 25) having two conformations, and 26.0% (*n* = 19) having three conformations (*Figure 6—figure supplement 1A*). Compared to the single-conformer deposited models, qFit-ligand improved the RSCC in 69.9% (*n* = 51) of structures (*Figure 6A*). We observed a correlation between the number of conformers generated by qFit-ligand and the RSCC of the input model (*Figure 6—figure supplement 1B*), where a lower input RSCC increases the likelihood of identifying more alternate conformers. For our EDIAm calculations, we assume that the electron density contribution from an atom adopts a spherical shape. This assumption becomes invalid at map resolutions worse than 2 Å, as the atomic scattering factors can no longer be reliably approximated by the Gaussian functions. Only 36 structures met the 2 Å resolution criteria, and we computed EDIAm values for this subset. Among them, 58.3% (*n* = 21/36) showed higher EDIAm values in the qFit-ligand models compared to their corresponding deposited structures (*Figure 6B*). Torsion strain analysis showed that 57.5% (*n* = 42/73) of structures had a lower qFit-ligand model strain, with a mean strain difference of –0.1 kcal/mol. This indicates that, on average, our

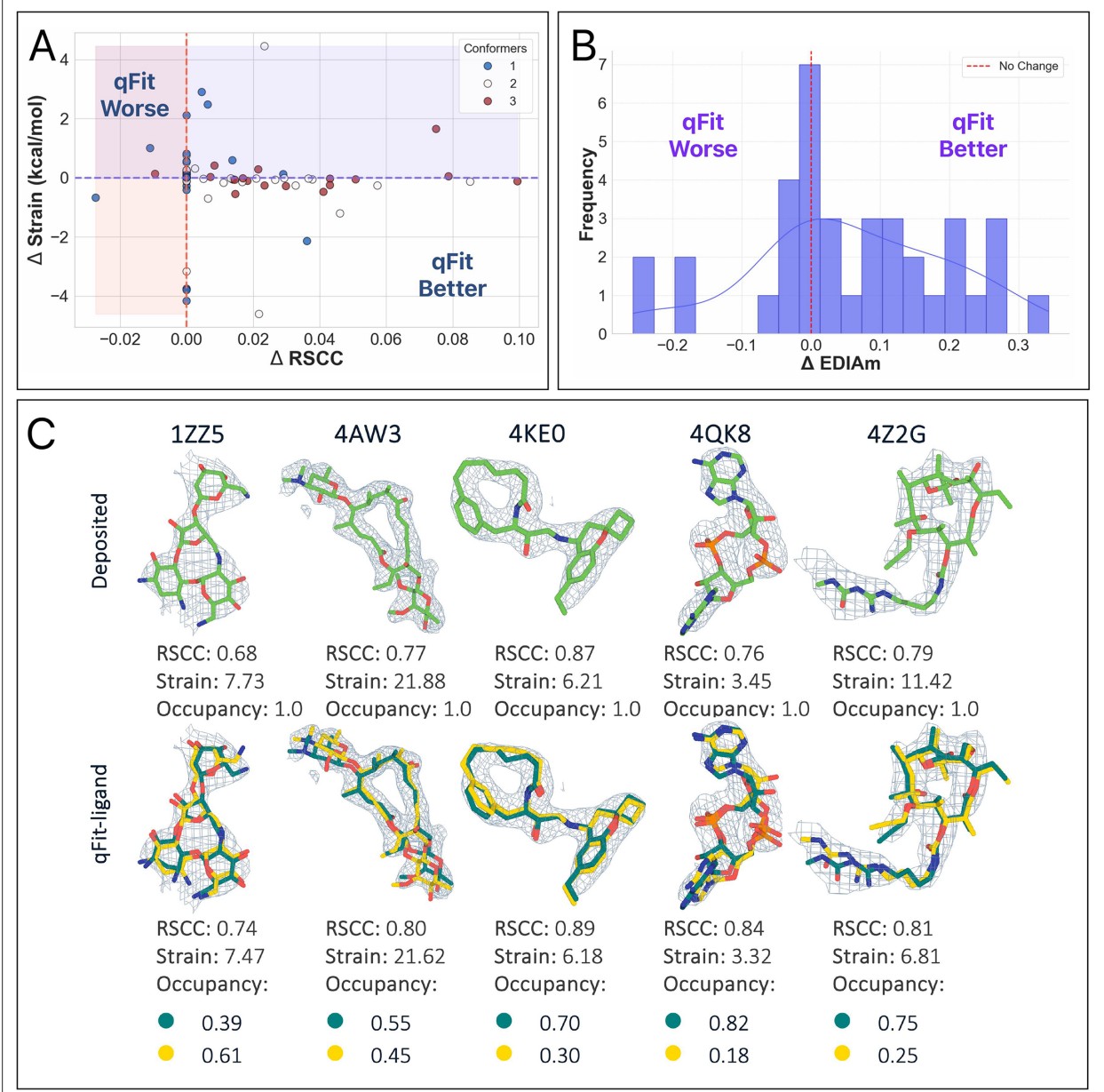

**Figure 6.** Evaluation of qFit-ligand predicted macrocycle conformations. (**A**) Differences in real space correlation coefficients (RSCC) and torsion strain between qFit-ligand predicted structures and refined deposited single-conformer macrocycles. The lower right quadrant shows structures for which we improve both RSCC and strain. (**B**) Differences in EDIAm values between the qFit-ligand and deposited models. Bars to the right of the vertical axis represent structures where the qFit-ligand model fits better to the electron density map. (**C**) Gallery of examples for which the qFit-ligand models have improved RSCC and strain compared to the deposited single-conformer macrocycle ligand. The composite omit density map is contoured at 1σ for every structure.

The online version of this article includes the following figure supplement(s) for figure 6:

**Figure supplement 1.** The number of macrocycle alternative conformers produced per PDB and their relationship to fit to density.

models maintain a similar level of energetic favorability as the deposited structures, while improving the fit to density (*Figure 6A, C*).

A few outlier cases have substantially reduced strain in the qFit-ligand models, particularly PDB 4Z2G, which shows a decrease of 4.61 kcal/mol (*Figure 6—figure supplement 1C*). In this case, qFit-ligand generated two conformers: one similar, including the strained pathologies, to the deposited model and a second, distinct conformer. Using COOT's ligand distortion tool, we compared the

strain between the deposited and this distinct qFit-ligand 'B' conformer by analyzing each bond and angle (*Emsley, 2017*). This tool evaluates deviations from ideal geometries based on COD (Crystallography Open Database) data, with restraint dictionaries generated through the AceDRG program (*Long et al., 2017b*; *Long et al., 2017a*) (Methods). In the qFit model, the overall strain is lower because alternative conformer 'A' is now at partial occupancy, and the 'B' conformer has much lower strain. Overall, while qFit-ligand primarily improves RSCC across most models, in a subset of cases, it also substantially reduces strain, demonstrating its ability to enhance both the fit and the energetic favorability of macrocycle conformations.

## qFit-ligand recovers heterogeneity in fragment-soaked event maps

X-ray crystallography-based fragment screens have taken off in academic and industry settings (*Günther et al., 2021*; *Gahbauer et al., 2023*; *Hartshorn et al., 2005*; *Badger, 2012*). Accurately modeling fragments is essential for effective building and merging strategies to create more drug-like molecules. However, as fragments are often bound at low occupancy, modeling into traditional $2F_o - F_c$ maps is incredibly difficult. To overcome this, 'event maps' are often created to detect low-occupancy ligands by averaging electron density across many apo datasets and subtracting these from the density of a potential ligand-bound structure (*Pearce et al., 2017*). This produces a ligand binding 'event map' and an estimate of the ligand occupancy. Once event maps are created, a modeler must manually fit the single or multiple conformations of the ligand into it. Therefore, we wanted to determine if qFit-ligand could automatically identify and model multiple conformations in event maps.

To assess qFit-ligand's ability to detect multiple conformations in event maps, we took advantage of ongoing fragment-based drug discovery efforts through the UCSF QCRG Antiviral Drug Discovery (AViDD) program to design inhibitors against the severe acute respiratory syndrome-coronavirus-2 NSP3 macrodomain (*Gahbauer et al., 2023*; *Suryawanshi et al., 2024*; *Correy et al., 2024*). We identified previously published and new fragments manually modeled with multiple conformations (*n* = 20) (*Supplementary file 5, table 5*). We used these as a true positive dataset to determine if we could identify multiple fragment conformations in event maps using qFit-ligand.

We created a modified true positive dataset (*n* = 20) by removing all 'B' conformers and setting the 'A' conformer occupancy to 1.0. qFit-ligand was then run as described above, but with an event map, rather than a composite omit map (Methods). To determine how precisely we captured the second conformation, we calculated the RMSD between the manually modeled 'B' conformer and the closest qFit-ligand conformer for each structure (*Figure 7A*). Only 9 of the structures exhibit an RMSD of less than 0.5 Å, indicating that for approximately half of the cases, our algorithm struggles to recapitulate the second deposited conformer. Of the 11 fragments with poor RMSD, about a third (*n* = 4/11) adopted a completely different binding pose, which our current algorithm often fails to capture accurately due to reliance on the input model. This highlights a limitation of our sampling strategy and suggests a potential direction for future development (*Figure 7—figure supplement 1*).

Despite this, compared to the modified true positive models, the qFit-ligand models had a higher RSCC in 17 structures and a higher EDIAm in 15 (*Figure 7B, C*). There are a number of structures for which we calculate an RMSD >0.5 Å and also an improved qFit-ligand map-to-model fit. In many of these cases, the improvement is generally very small. In others, we believe they represent situations where multiple combinations of conformations can accurately represent the underlying data. For instance, in PDB 7HHW, the qFit-ligand model generated a flipped Thiophene compared to the deposited model, resulting in a relatively high RMSD to the deposited 'B' while still providing an equally good fit to the electron density (*Figure 7—figure supplement 2*). In addition, the torsion strain analysis reveals that 13 structures have a lower strain in the qFit-ligand model, and 12 structures have both a higher RSCC and a lower model strain in the qFit-ligand model (*Figure 7B*). While the mean strain improvement in the qFit-ligand models was marginal, only –0.6 kcal/mol, it indicates that we can reliably fit to density without straining the molecule (*Figure 7D*).

In this use case, qFit-ligand models alternative conformations into an event map, which represents only partial occupancy of the unit cell. Therefore, we scale the output ligand conformer occupancies to estimated occupancy from the background density correction prior to merging into the full system. Following this scaling, we perform standard refinement and note that the sum of occupancy across ligand conformations is a refined variable that can be <1. Together, these results suggest that

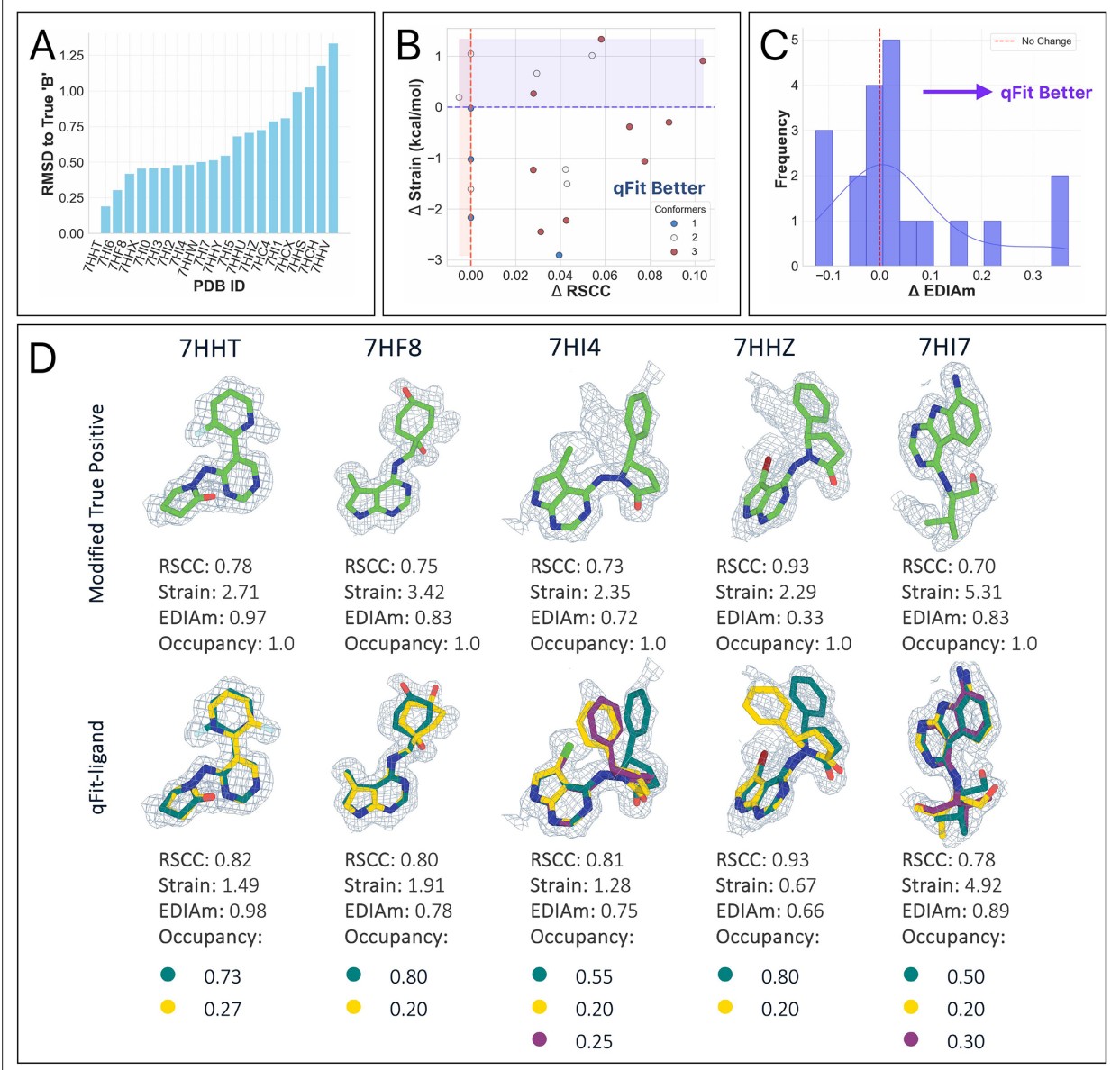

**Figure 7.** Evaluation of qFit-ligand on fragments in PanDDA maps. (**A**) Root mean square deviation (RMSD) between the deposited 'B' conformer and the closest qFit-ligand conformer. Lower values correlate with a closer recapitulation of the deposited heterogeneity. (**B**) Real space correlation coefficients (RSCC) and torsion strain differences in the deposited models and the qFit-ligand predicted models. The lower right quadrant shows structures for which we improve both RSCC and strain. (**C**) Differences in EDIAm values between the qFit-ligand and modified true positive models. Bars to the right of the vertical axis represent structures where the qFit-ligand model fits better to the event map. (**D**) Gallery of examples for which qFit-ligand successfully recovers well-fitting alternate conformers. The composite omit density map is contoured at 1σ for every fragment.

The online version of this article includes the following figure supplement(s) for figure 7:

**Figure supplement 1.** PDB 7HHU represents the structure with the highest root mean square deviation (RMSD) between its deposited 'A' (green) and 'B' (gray) conformers.

**Figure supplement 2.** Comparison of deposited conformers 'A' (green), 'B' (gray), and qFit-ligand conformers for PDB 7HHW.

qFit-ligand can be used alongside manual modeling of fragment modeling, but additional development is needed where there are large ligand conformational changes.

## qFit-ligand models multiple conformations of ligands into cryo-EM density maps

Recent advances in cryo-EM are resulting in many reconstructions with better than 2 Å resolution. At this resolution, it is possible to resolve conformational heterogeneity at the atomic level, prompting us to determine if qFit-ligand can capture this heterogeneity. To evaluate the performance of qFit-ligand on cryo-EM data, we examined recently deposited inhibitors of human CDK-activating kinase, a three-subunit protein complex recognized as a compelling candidate for cancer and antiviral drug development (*Cushing et al., 2024*). Four of their molecules were manually modeled as multiconformer ligands, providing a valuable set of true positives (*Figure 8*). All map resolutions were better than 2 Å, and ligand molecular weights were between 350 and 397 Da (*Supplementary file 7, table 7*).

For each deposited model, we created a new modified single-conformer true positive following the same procedure as outlined above. Next, we applied a similar pipeline as with the other true positive datasets. For both refinement stages, we instead used *phenix.real_space_refine* (Methods) with the deposited EM map. In all four structures, qFit-ligand identified a second conformation. Compared to the modified single-conformer true positive input model, all generated multiconformers were improved in terms of RSCC, EDIAm, and strain (*Figure 8*). Across the four structures, RSCC increased by up to 0.1, EDIAm by up to 0.2, and strain decreased by up to 2.3 kcal/mol in the qFit-ligand models. Interestingly, in all of the structures, qFit-ligand placed slightly different conformations and occupancies compared to the deposited model, however, all resulted in improved RSCC and strain, although two out of the four structures had lower EDIAm (*Figure 8*). This result suggests that qFit-ligand can be used to model conformational heterogeneity to improve model quality in cryo-EM derived structures, although this needs to be examined with a larger dataset.

## Discussion

Although ligands can retain conformational flexibility when bound to receptors, this conformational heterogeneity is rarely captured in deposited models, potentially leading to misinterpretations of protein–ligand interactions (*van Zundert et al., 2018*; *Skaist Mehlman et al., 2023*; *Zhou and Hong, 2021*). A key reason for this modeling gap is the significant compositional and conformational heterogeneity surrounding ligands, making accurate modeling of ligands particularly challenging. qFit-ligand directly tackles the challenge of conformational heterogeneity by automatically modeling alternative ligand conformations in high-resolution X-ray crystallography and cryo-EM maps with clear unmodeled features. It is advisable to employ qFit-ligand selectively, focusing on cases where there is a moderate correlation between the input model and the experimental data, strong visual density in the binding pocket, high map resolution, or when a single-conformer ligand model is strained.

The major advancements in qFit-ligand presented here stem from integrating the torsionally aware sampling strategy from RDKit, resulting in a reduction of ligand torsional strain, while still providing an increase in model to map fit. We observed this improvement in both ligands that fit by multiple low-strain conformations, along with improved fit of single ligand conformations. High ligand strain is energetically unfavorable and can reduce binding affinity (*Sitzmann et al., 2012*). Therefore, we would expect that most observed protein–ligand complexes in the PDB are likely to represent relatively unstrained ligands. While the majority of deposited structures are low in strain, analyses across multiple tools reveal a wide distribution of strain observed, with many structures having high strain (*Smola et al., 2021*; *Jain et al., 2023*; *Tong and Zhao, 2021*). We demonstrate the possibility of using qFit-ligand to fix high-strain ligands by identifying where multiple ligand conformations should be used, or by improving the geometry of a single conformation through torsionally aware sampling.

In addition to these overall improvements, we have expanded qFit-ligand in three primary directions. First, we enabled qFit-ligand to model conformational heterogeneity in macrocycles. Macrocycles have the possibility for targeting 'undruggable' proteins because their exceptional conformational flexibility allows them to interact effectively with relatively flat protein surfaces (*Vinogradov et al., 2019*). We showed that qFit-ligand can parsimoniously capture the heterogeneity present in bound macrocycles, often improving fit to density while lowering strain compared to the deposited

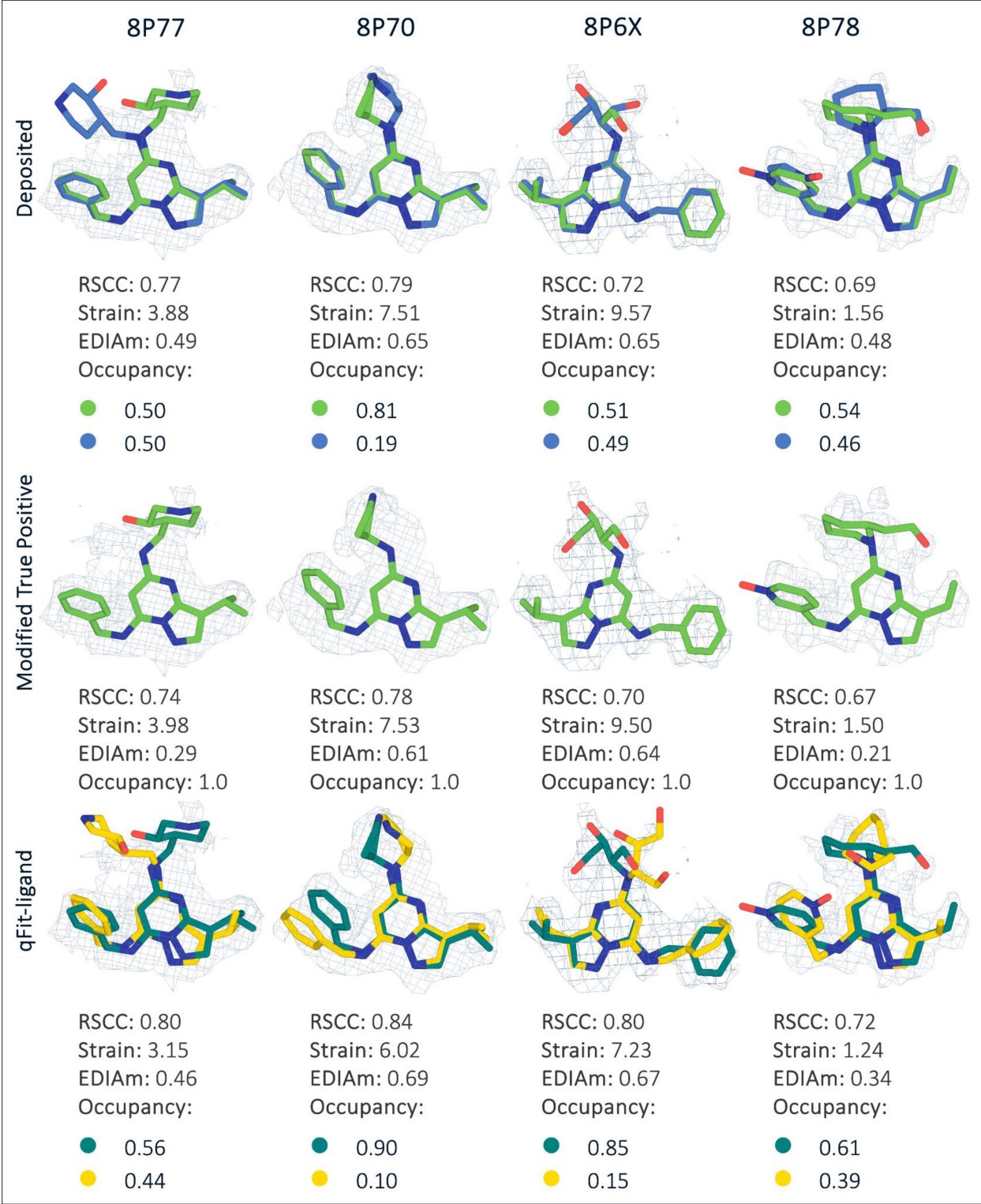

**Figure 8.** Gallery of the four cryo-electron microscopy (cryo-EM) structures with deposited model, modified true positive, and qFit-ligand structure. In each case, the qFit-ligand model outperforms the modified true positive model in all validation metrics. The EDM density map is contoured at 1σ for every structure.

single-conformer structures. While a previous effort, XGen, showed that ensemble representations of macrocycles also reduced the strain compared to deposited structures (*Jain et al., 2020*), ensemble models are complex to analyze, difficult to manipulate in model building software (*Emsley et al., 2010*), and require specific refinement protocols that prevent easy integration into modeling pipelines.

Second, we have added capabilities to model partially occupied fragments from high-throughput screening campaigns. The popularity of X-ray crystallography-based fragment screening has surged due to beamline improvements and algorithmic developments that enhance detection of low-occupancy binding events (*Pearce et al., 2017*). Fragments can potentially bind in multiple conformations due to the small size and promiscuous or weak interactions (*Bian and Xie, 2018*). We expanded qFit-ligand to automatically model fragments into event maps. Because of the weak signal in event maps, we emphasize the importance of manual scrutiny of the output conformations for fragments to an even greater extent than for larger, fully occupied ligands. Further, we identified that qFit-ligand has difficulty identifying alternative conformations resulting from larger translations and 'ligand flips' that are more common with fragment screening. We have also added an experimental flag that samples 180° flips of ligands; however, this approach should only be used as an exploratory tool where there is a strong visual prior.

Third, we can now apply qFit-ligand to cryo-EM data. This capability opens up exciting new opportunities for structure-based drug discovery. There are several applications where cryo-EM is better suited to experimental objectives than crystallography, including the characterization of dynamic and heterogeneous biomolecular assemblies in more native-like environments (*Wang and Wang, 2017*). As a proof of principle, we showed that qFit-ligand can recapitulate deposited alternate conformations from a ligand series, demonstrating its potential to model multiconformer ligands directly into high-resolution cryo-EM density maps.

Despite these advancements, qFit-ligand has room for further improvement. Our approach is limited by its reliance on an initial single-conformer structure, which introduces bias toward the starting model and hinders effective exploration of conformational space when the input ligand is poorly resolved. Additionally, qFit-ligand performs best when unmodeled density is consistent with subtle conformational changes, such as torsion angle variations or minor translational shifts. It struggles to identify more dramatic conformational heterogeneity, such as 180° flip ligand conformations as seen in our fragment dataset (*Gahbauer et al., 2023*; *Suryawanshi et al., 2024*; *Correy et al., 2024*). As such, qFit-ligand primarily serves as a 'thought partner' for manual modeling. Modelers still must resolve many ambiguities, including initial ligand placement, in order to fully take advantage of qFit capabilities. In active modeling workflows or large-scale analyses, the workflow would only accept the output of qFit-ligand when it improves model quality. In cases where qFit-ligand degrades map-to-model fit and/or strain, we can simply revert to the input model. In practice, users can easily remove poorly fitting conformations using molecular modeling software such as COOT, while keeping the well-modeled conformations, which is an advantage of the multiconformer approach over ensemble refinement methods.

Additionally, our algorithm's placement within the larger refinement and ligand modeling ecosystem highlighted other areas that need improvement. We note that macrocycles, due to their complicated and interconnected degrees of freedom, suffer acutely from the refinement issues, as demonstrated by the failure of approximately one-third of datasets in our standard preparation or post-refinement pipelines due to ligand parameterization issues. Many of these stemmed from problematic ligand restraint files, highlighting the difficulty of encoding the geometric constraints of macrocycles using standard restraint libraries. Improved force fields or restraints for macrocycles are desperately needed to improve their modeling. New approaches such as quantum mechanical restraints refinement (*Liebschner et al., 2023*), which replaces standard geometric restraints with in situ energy-minimized quantum calculations, may offer a path toward more accurate modeling of chemically complex ligands. We note that even linear non-canonical peptides present similar failure modes to macrocycles, with a mix of ATOM and HETATM records and the need for custom cif definitions and link records. For these reasons, we did not include analysis on small peptide ligands; however, canonical peptides can be modeled with standard qFit (*Wankowicz et al., 2024c*).

Finally, we ultimately strive for modeling the conformational heterogeneity across the entire system including ligands, proteins, nucleic acids, and water molecules. Currently, qFit algorithms allow for modeling either the protein or the ligand separately, focusing on the conformational possibilities of

one while treating the other as static (**Wankowicz et al., 2024c**). Joint modeling across all system components would generate conformational ensembles that enhance our understanding of how the conformational heterogeneity of each component impacts the other. However, the combinatorial complexity of such problems is ill-suited for the sample-and-select strategies employed by qFit. Beyond the computational modeling advancements, without machine-readable and human-interpretable encoding, we will remain limited in understanding the natural heterogeneity that impacts molecular recognition and drug design (**Wankowicz and Fraser, 2024b**). Overall, qFit-ligand provides structural biologists with an efficient tool for modeling parsimonious multiconformer ligand models that fit optimally into electron density maps, reducing the need for manual intervention, aiding in understanding how conformational heterogeneity impacts ligand binding and downstream biology.

## Methods
### Running qFit-ligand
SMILES strings used as input for qFit-ligand are fetched from the PDB, given the three-letter ligand identifier. Our RDKit-based conformer sampling is initialized from the input PDB file; however, RDKit often misassigns bond orders when interpreting PDBs directly. Therefore, we use the SMILES string as a template for correcting bond orders in the generated conformers.

To run qFit-ligand on regular small molecules and macrocycles, we used the following command:

qfit_ligand composite_omit_map.mtz refined_pdb.pdb -sm<smiles string> -l 2FOFCWT,PH-2FOFCWT <chain,res_num> –p 5.

To run qFit-ligand when using an event map, we used the following command:

qfit_ligand event_map.ccp4 input_model.pdb -sm<smiles string> -l FWT,PHWT -r<resolution > <chain,res_num> –p 5

To run qFit-ligand when using a cryo-EM map, we used the following command:

qfit_ligand <emd_map>.map input_model.pdb -sm<smiles string> -r<resolution > <chain,res_num> –cryo_em_ligand –p 5

Code for running qFit-ligand is available in our Github repository (https://github.com/ExcitedStates/qfit-3.0 copy archived at **Riley et al., 2025**) under version 2024.3 and SBGrid (https://sbgrid.org/).

### RDKit's ETKDG implementation
ETKDG is an enhancement of traditional Distance Geometry (DG), implemented within RDKit's EmbedMultipleConfs function (**Riniker and Landrum, 2015**). During the distance bounds matrix construction, bounds are set for 1–2 (bonded atoms), 1–3 (bond angle related atoms), 1–4 (torsion angle related atoms), and 1–5 interactions, based on empirical knowledge of ideal bond lengths and angles from chemical structures. These bounds are subsequently sampled and embedded into 3D coordinates. Next, a minimization step is performed using SMARTS patterns to identify torsional substructures in the molecule (**Schärfer et al., 2013**), where for each SMARTS identified torsion pattern, the corresponding torsional potential is applied to the sampled conformation. These energy functions describe the energetic preference for specific dihedral angles and guide the RDKit-generated torsions toward experimentally observed angle ranges. The functional form is expressed as a cosine series expansion, and the parameters are fit to experimental torsion angle distributions from the CSD. Following the torsional minimization, we apply a force field minimization using the MMFF94 force field via the ForceField.rdForceField module of RDKit. The force field has the functional form of

$$E_{MMFF} = \sum EB_{ij} + \sum EA_{ijk} \sum EBA_{ijk} + \sum EOOP_{ijk;l} + \sum ET_{ijkl} + \sum EcdW_{ij} + \sum EQ_{ij}$$

Where the terms refer to bond stretching, angle bending, stretch-bend, out-of-plane bending, torsional, van der Waals, and electrostatic, respectively (**Tosco et al., 2014**).

### Pre-qFit refinement protocol
#### For X-ray maps
Before running qFit-ligand, all input models are stripped of their alt confs, resulting in a set of single-conformer coordinate files with 'A' ligand occupancies set to 1.0. We use phenix.ready_set (or phenix.

elbow if phenix.ready_set fails) to generate cif files for ligand restraint during refinement. All pre-qFit refinement uses the following parameters.

```
refinement.refine.strategy=individual_sites+individual_adp+occupancies
refinement.input.monomers.file_name=ligand.cif
refinement.main.number_of_macro_cycles=5
refinement.main.nqh_flips=True
refinement.output.write_maps=False
refinement.hydrogens.refine=riding
refinement.main.ordered_solvent=True
refinement.target_weights.optimize_xyz_weight=True
refinement.target_weights.optimize_adp_weight=True
refinement.input.xray_data.r_free_flags.generate=True
```

After refinement, we generate a composite omit map from the refined model to use as qFit-ligand input.

```
phenix.composite_omit_map refined_model.pdb data.mtz omit-type=refine nproc=8r_free_
flags.generate=True exclude_bulk_solvent=True
```

Setting *exclude_bulk_solvent=True* prevents the bulk solvent model from being applied, which typically accounts for disordered solvent by filling low-density areas in the map. When bulk solvent correction is included, it adjusts the electron density by assuming the presence of uniform solvent in regions of low density, such as areas surrounding the ligand. This can reduce the contrast between weak ligand density and the surrounding solvent, potentially smearing or flattening the electron density around flexible or poorly ordered regions like alternative ligand conformations. By excluding bulk solvent correction, you retain the raw electron density in those regions, ensuring the density is not artificially raised or smoothed. This allows clearer visualization of weak or partial densities that might indicate alternative conformers.

## For cryo-EM maps

For cryo-EM data, we use a similar refinement protocol, but instead use phenix.real_space_refine (*Afonine et al., 2018*). We mainly use default parameters, but specify the following:

```
refinement.macro_cycles=5
pdb_interpretation.apply_cif_restraints.restraints_file_name
```

## Post-qFit refinement protocol

After qFit-ligand is run, and before the final refinement, if there are any conformers <0.1 occupancy, they are culled from the output multiconformer model. Again, we use phenix.ready_set (or phenix. elbow if phenix.ready_set fails) to generate cif files for ligand restraint during refinement. All crystal structures are subsequently refined with the following parameters.

```
refinement.refine.strategy=individual_sites+individual_adp +occupancies
refinement.input.monomers.file_name=ligand.cif
refinement.main.number_of_macro_cycles=5
refinement.main.nqh_flips=True
refinement.refine.adp.individual.isotropic=all
refinement.output.write_maps=False
refinement.hydrogens.refine=riding
refinement.main.ordered_solvent=True
refinement.target_weights.optimize_xyz_weight=True
refinement.target_weights.optimize_adp_weight=True
```

All cryo-EM structures are subsequently refined with default parameters along with specifying the following parameters:

```
refinement.macro_cycles=5
pdb_interpretation.apply_cif_restraints.restraints_file_name
```

After five macro cycles of refinement, we then remove and redistribute the occupancy of any conformers with less than 10% occupancy. We do not re-refine after this redistribution.

If running qFit-ligand on an event map, the refinement process involves an additional step. When using the optional --BDC flag, the script scales the occupancies of the qFit-ligand generated conformers by a factor of (1 − BDC), and produces a new protein-ligand PDB file with the adjusted occupancies. The new PDB file is then processed through the standard refinement protocol, as described above.

### Ligand geometry validation of macrocycles

To validate the geometry of the macrocyclic ligands, we employed a quick check to ensure that the chemical structure had not been altered during refinement. Specifically, we checked that the chemical connectivity of the ligand remained unchanged, even if the conformation varied.

1. Load the PDB file of the protein–ligand complex along with the SMILES string of the bound ligand. The SMILES string represents the correct chemical connectivity of the ligand as it should appear post-refinement.
2. Use RDKit to interpret the SMILES string and attempt to assign bond orders to the ligand in the PDB file. This step compares the intended chemical structure (from the SMILES) with the actual structure after refinement. The bond order assignment is used as a proxy to check if the refinement process altered the ligand's chemical connectivity.
3. If RDKit successfully assigns bond orders, it suggests that the chemical connectivity has been preserved, and that the refinement process did not improperly modify the ligand's geometry. However, if RDKit encounters difficulties assigning bond orders, this signals that the refinement may have detrimentally altered the ligand's structure.

This method serves as a fast, automated 'sanity check' to flag potential problems, helping to avoid the need for manual inspection of each PDB file.

### COOT's ligand distortion tool

To examine how conformational differences impacted strain in select examples from the macrocycle dataset, we used COOT's ligand distortion tool (*Emsley, 2017*). The penalty score is calculated using Hooke's Law, where target values and sigma values from the restraint files are used. The non-bonded interactions are penalized using the Lennard–Jones potential, with atom radii taken from the CCP4 geometry tables. Larger deviations from ideal geometries result in higher penalties, and the overall penalty score is calculated as $\left(\frac{deviation}{\sigma}\right)^2$, where σ represents the standard deviation of the target value, functioning as the spring constant in Hooke's Law.

### Scoring

QP solvers handle Quadratic Programming problems (*Agrawal et al., 2017*; *Diamond and Boyd, 2016*). These problems involve an objective function that is quadratic (a polynomial of degree two) and is subject to linear constraints. The primary goal in the QP framework is to find the combination of conformer occupancies, stored in vector $\omega = <\omega_0, ..., \omega_n>$, that minimize the difference between the observed electron density and the electron density calculated from the model. Mathematically, this minimizes a residual sum-of-squares function, $rss(\omega)$:

$$min_\omega(rss(\omega)) = min_\omega(\| \rho^c \omega - \rho^o \|^2)$$

$\rho^o$ is the observed electron density from the user-provided map (target)
$\rho^c$ is the weighted calculated electron density from conformers

These occupancies are meaningful parameters, so we require that their sum is within the unit interval, ensuring the total model density does not surpass 100% occupancy.

$$\Sigma\omega_i \leq 1$$

Each individual occupancy must be a positive fractional number, meaning each conformer's contribution is between none and full.

$$0 \leq \omega_i \leq 1$$

MIQP solvers extend the capabilities of QP solvers by incorporating integer constraints into the optimization problem.

Again, we set up the minimization problem:

$$min_\omega(rss(\omega)) = min_\omega(\| \rho^c \omega - \rho^o \|^2)$$

$$\Sigma \omega_i \leq 1$$

Here, we select up to a predetermined number of conformers (cardinality) that meets a minimum occupancy threshold, with all others set to zero. This selection is achieved through mixed-integer linear constraints:

$$z_i t_{min} \leq \omega_i \leq z_i$$

where

$$z_i \in \{0, 1\}$$

$t_{min}$ is the minimum-allowable occupancy value for $\omega_i$. If $\omega_i$ in non-zero, it must be at least $t_{min}$.

The integer constraint limits the number of conformers explicitly. Cardinality is set to three, and the minimum occupancy $t_{min}$ set to 0.20, so only up to three conformers can have non-zero weights (of at least $t_{min}$) in the final multiconformer model.

Should the user include the optional '--cryo_em_ligand' flag on the command line, the cardinality will be reduced from three to two.

## RSCC

The RSCC is a metric used to assess how well a modeled structure fits into the observed electron density in a crystallographic experiment. It compares the observed electron density values with the electron density values calculated from the model. RSCC values range from 0 to 1, with values above 0.80 generally indicating a good fit. RSCC is calculated using a linear sample correlation coefficient formula:

$$RSCC = coor(\rho_{obs}, \rho_{calc}) = \frac{cov(\rho_{obs}, \rho_{calc})}{\sqrt{var(\rho_{obs})var(\rho_{calc})}}$$

$$= \frac{\sum |\rho_{obs} - <\rho_{obs}>| \sum |\rho_{calc} - <\rho_{calc}>|}{\sqrt{\sum |\rho_{obs} - <\rho_{obs}>|^2 \sum |\rho_{calc} - <\rho_{calc}>|^2}}$$

where $\rho_{obs}$ is the observed electron density at grid points covering the residue of interest (the input density map), and $\rho_{calc}$ is the density map calculated from the model (*Tickle, 2012*).

To calculate RSCC, we must first determine which density map voxels belong to the ligand. We created a mask around the coordinates of the full qFit-ligand ensemble, and only the density values under this mask's footprint are extracted for the calculation. The same mask is used to calculate the RSCC of the input (single-conformer) model versus the qFit-ligand model.

Code for calculating RSCC is available on our GitHub repository.

## EDIAm

EDIA is a method for estimating the electron density support for an individual atom in a density map. This is determined by sampling grid points $p$ in a sphere around the atom of interest $a$, and calculating the weighting factor, an ownership value, and the density score (*Meyder et al., 2017*).

$$\text{EDIA}(a) = \frac{\sum\limits_{p \in M_{2f0-fc}} w(p,a)\, o\,(p,a) z(p)}{\sum\limits_{p \in M_{2f0-fc | w(p,a) > 0}} w(p,a)}$$

The distance-dependent weighting factor $\omega(p, a)$ distinguishes between meaningful and excess electron density near atom $a$, assigning negative weights to density located outside the atom's expected region. The ownership function $o(p, a)$ allocates each grid point $p$ to one or more atoms, determining which parts of the map are attributed to which atoms. The density score $z(p)$ for an atom $a$ is then computed as follows:

$$z(p) = \begin{cases} 0 & \text{if } \dfrac{\rho(p) - \mu}{\sigma} < 0.0 \\ \dfrac{\rho(p) - \mu}{\sigma} & \text{if } 0 \leq \dfrac{\rho(p) - \mu}{\sigma} \leq \zeta \\ \zeta & \text{if } \dfrac{\rho(p) - \mu}{\sigma} > \zeta \end{cases}$$

where $\zeta = 1.2$, $\rho(p)$ represents the density at grid point $p$, $\mu$ is the mean of the $2_{fo-fc}$ map, and $\sigma$ is the root mean square of the $2_{fo-fc}$ map. To quantify the fit of an entire molecule to the electron density map (EDIAm), the EDIA score is first computed for each atom individually and then combined across all atoms in the ligand. Code for calculating EDIAm is available on our GitHub repository.

## RMSD

RMSD is a widely used metric in structural biology for comparing molecular conformations. It measures the average distance between corresponding atoms of two superimposed structures and is valuable for assessing differences in conformers, protein structures, and ligand poses.

The RMSD between two sets of atomic coordinates is calculated using the formula:

$$RMSD = \sqrt{\frac{1}{N} \sum_{i=1}^{N} [(x_i^{(1)} + x_i^{(2)}) + (y_i^{(1)} - y_i^{(2)}) + (z_i^{(1)} - z_i^{(2)})]}$$

where $N$ is the number of atoms, and $\left(x_i^{(1)}, y_i^{(1)}, z_i^{(1)}\right)$ and $\left(x_i^{(2)}, y_i^{(2)}, z_i^{(2)}\right)$ are the coordinates of the $i$th atom in the two conformers.

Code for calculating the RMSD between two conformers of a ligand is available on our GitHub repository.

## Torsion strain

To calculate molecular strain, we take advantage of software available at https://tldr.docking.org/ **Gu et al., 2021**.

The TLDR software employs a statistical method based on torsion patterns observed in crystal structures. It identifies all torsions in an input molecule, where each pattern consists of a sequence of four atoms forming a dihedral angle. These patterns are compared against a pre-compiled library of torsion energies sourced from the CSD and PDB.

For each torsion pattern, the software retrieves a histogram of observed dihedral angles and their associated energies. The dihedral angle of the molecule's conformation is matched to this histogram, and the corresponding energy is determined. This process is repeated for all torsion patterns in the molecule, and the total strain energy is calculated by summing the individual torsion energies.

## Generating a synthetic dataset

To create our synthetic dataset, we constructed four multiconformer ligands using COOT (**Emsley et al., 2010**). We generated five new PDB files for each ligand, varying the occupancy between the two conformers in the ratios: 0.50/0.50, 0.40/0.60, 0.30/0.70, 0.20/0.80, and 0.10/0.90. These files represent different relative populations of the conformers. For each of these ligand models, we produced a series of electron density maps covering resolutions from 0.8 to 2.5 Å, with increments of 0.1 Å using *phenix.fmodel*. This process involves the following steps.

For each given ligand input coordinate file, the script adjusts the B-factors, or temperature factors, of ligand atoms based on the specified resolution. As the resolution degrades from 0.8 to 2.5 Å, the B-factors incrementally increase. This adjustment models the increased positional uncertainty of atoms that typically occurs at lower resolutions. The modified ligand structures with these adjusted B-factors at each resolution level are saved as new PDB files. Following this, the script utilizes *phenix.fmodel* to calculate theoretical structure factors from each altered atomic model. These structure factors are then used to compute synthetic electron density maps. To each of these maps, we generate and add random Gaussian noise values scaled proportionally to the resolution. This scaling reflects the escalation of experimental noise as resolution deteriorates, a common occurrence in real-life crystallographic data.

*phenix.fmodel* is used with the following parameters:

```
phenix.fmodel input_pdb_file.pdb k_sol=0.4 b_sol=45 high_resolution=<resolution > r_free_
flags_fraction=0.05 output.file_name = output_file.mtz.
```

The full script is available at: https://github.com/fraser-lab/qFit_biological_testset (copy archived at *Ravikumar and Wankowicz, 2024*).

## X-ray crystallography

Mac1 crystals (P43 construct, residues 3–169) were grown by sitting-drop vapor diffusion in 28% wt/vol 570 polyethylene glycol 3000 and 100 mM *N*-cyclohexyl-2-aminoethanesulfonic acid pH 9.5 as described previously (*Gahbauer et al., 2023*; *Schuller et al., 2021*). Compounds prepared in DMSO (100 mM) were added to crystal drops using an Echo 650 acoustic dispenser (final concentration of 10 mM) (*Collins et al., 2017*). Crystals were incubated at room temperature for 2–4 hr prior to vitrification in liquid nitrogen without additional cryoprotection. X-ray diffraction data were collected at the Advanced Light Source (ALS beamline 8.3.1) or the Stanford Synchrotron Light Source (SSRL beamline 9–2). Data were indexed, integrated, and scaled with XDS (*Kabsch, 2010*) and merged with Aimless (*Evans and Murshudov, 2013*). The P43 Mac1 crystals contain two copies of the protein in the asymmetric unit (chains A and B). The active site of chain A is open; however chain B is blocked by a crystal contact. We previously observed that potent Mac1 inhibitors dissolve crystals, likely through the displacement of the B chain crystal contact (*Gahbauer et al., 2023*). In addition, crystal packing in the chain A active site restricts movement of the Ala129–Gly134 loop, leading to decreased occupancy for compounds with substituents on the pyrrolidinone. To aid modeling the resulting conformational and compositional disorder, we used the PanDDA method (*Pearce et al., 2017*) to model ligands where the occupancy was low (<25%) or where there was substantial disorder. After modeling ligands, structures were refined using phenix.refine (*Liebschner et al., 2019*) as described previously (*Gahbauer et al., 2023*). Data collection settings and statistics are reported in *Supplementary file 6, table 6*.

## Chemical synthesis

Unless otherwise noted, all chemical reagents and solvents used are commercially available. Air and/or moisture-sensitive reactions were carried out under an argon atmosphere in oven-dried glassware using anhydrous solvents from commercial suppliers. Air and/or moisture-sensitive reagents were transferred via syringe or cannula and were introduced into reaction vessels through rubber septa. Solvent removal was accomplished with a rotary evaporator at ca. 10–50 Torr. Microwave reactions were carried out in a CEM Discover microwave reactor. Chromatography was carried out using the Isolera Four flash chromatography system with Silia*Sep* silica gel cartridges from Silicycle.

Reverse phase chromatography was carried out on

1. Waters 2535 Separation module with Waters 2998 Photodiode Array Detector. Separations were carried out on XBridge Preparative C18, 19 × 50 mm column at ambient temperature
2. Gilson GX-281 instrument column: Xtimate Prep C18, 21.2 × 250 mm, 150 Å, 10 μm particle size.

LC/MS data were acquired on

1. Waters Acquity UPLC QDa mass spectrometer equipped with Quaternary Solvent Manager, Photodiode Array Detector, and Evaporative Light Scattering Detector. Separations were carried out with Acquity UPLCÒ BEH C18 1.7 mm, 2.1 × 50 mm column at 25°C, using a mobile phase of water-acetonitrile containing a constant 0.1% formic acid.
2. Agilent 1200 Infinity LC with an Agilent 1956 single quadrupole MS using electrospray ionization: Column: SunFire C18 (4.6 × 50 mm, 3.5 um), Mobile phase: $H_2O$ (10 mmol $NH_4HCO_3$) (A) / ACN (B), Elution program: Gradient from 10 to 95% of B in 1.5 min at 1.8 ml/min, Temperature: 50°C, Detection: UV (214, 254 nm) and MS (ESI, POS mode, 103–100 amu).

Chemical shifts are reported in d units (ppm). NMR spectra were referenced relative to residual NMR solvent peaks. Coupling constants (*J*) are reported in hertz (Hz). NMR spectra were recorded on Bruker Avance III HD 400 MHz spectrometer or Bruker 500 MHz spectrometer.

## 4-Chloro-9*H*-pyrimido[4,5-*b*]indol-8-amine

A solution of 3-fluoro-2-nitroaniline (11 g, 70.51 mmol) in acetic anhydride (20 ml) was stirred at room temperature for 16 hr. The reaction mixture was filtered, and the solids were washed with petroleum ether (100 ml) and dried to obtain 10.7 g (77%) of *N*-(3-fluoro-2-nitrophenyl)acetamide as a brown solid. LC–MS (ESI): *m/z* = 199.3 (M+H)⁺.

To a solution of *N*-(3-fluoro-2-nitrophenyl)acetamide (10.7 g, 54.04 mmol) in DMF (100 ml) was added methyl 2-isocyanoacetate (8.02 g, 81.06 mmol) and potassium carbonate (14.92 g, 108.08 mmol). After stirring at 80°C for 2 hr, the reaction mixture was cooled to room temperature, acidified with 2 N HCl (ca. 2000 ml), and extracted with ethyl acetate (300 ml *3). The combined organic layers were washed with brine (100 ml), dried over sodium sulfate, and concentrated under reduced pressure. The residue was purified by silica gel chromatography (10:1 petroleum ether/ethyl acetate) to obtain 11 g (73%) of methyl 2-(3-acetamido-2-nitrophenyl)-2-isocyanoacetate as a yellow solid. LC–MS (ESI): *m/z* = 278.2 (M+H)⁺.

To a solution of methyl 2-(3-acetamido-2-nitrophenyl)-2-isocyanoacetate (11 g, 39.71 mmol) in *glacial* acetic acid (100 ml), was added slowly zinc dust (25.81 g, 397.10 mmol) in two portions. After stirring at 60°C for 2 hr, the reaction mixture was cooled to room temperature, filtered, and washed with THF. The filtrate was concentrated under reduced pressure and purified by silica gel chromatography (10:1 dichloromethane/methanol) to obtain 6.2 g (63%) of methyl 7-acetamido-2-amino-1*H*-indole-3-carboxylate as a yellow solid. LC–MS (ESI): *m/z* = 248.3 (M+H)⁺.

A solution of methyl 7-acetamido-2-amino-1*H*-indole-3-carboxylate (6.2 g, 25.10 mmol) in formamide (450 ml) was stirred at 220°C for 2 hr. The reaction mixture was then cooled to room temperature and poured in 100 ml of water. The resulting mixture was allowed to stand for 15 min before the solids were collected by filtration, washed with water, and dried to obtain 4.1 g of a 1:2 mixture of *N*-(4-hydroxy-9H-pyrimido[4,5-*b*]indol-8-yl)acetamide and *N*-(4-hydroxy-9*H*-pyrimido[4,5-*b*]indol-8-yl) formamide. This mixture was taken in methanol (25 ml) and aqueous 12 N NaOH (25 ml). After stirring at 60°C for 16 hr, the reaction mixture was then cooled to room temperature, concentrated under reduced pressure to remove methanol, and the residue was poured into 100 ml of water. The resulting mixture was allowed to stand for 15 min before the solids were collected by filtration, washed with water, and dried to obtain 3.5 g (70%) of 8-amino-9*H*-pyrimido[4,5-*b*]indol-4-ol as a brown solid. LC–MS (ESI): *m/z* = 201.2 (M+H)⁺.

A solution of 8-amino-9H-pyrimido[4,5-*b*]indol-4-ol (3.5 g, 17.5 mmol) in formamide (30 ml) was stirred at 150°C. After 6 hr, the reaction mixture was cooled to room temperature and poured into water (200 ml). The resulting mixture was allowed to stand for 15 min before the solids were collected by filtration, washed with water, and dried to obtain 3.5 g (88%) of *N*-(4-hydroxy-9*H*-pyrimido[4,5-*b*] indol-8-yl)formamide as a brown solid. LC–MS (ESI): *m/z* = 229.2 (M+H)⁺.

To a solution of *N*-(4-hydroxy-9*H*-pyrimido[4,5-*b*]indol-8-yl)formamide (3.5 g, 15.35 mmol) in phosphorus oxychloride (30 ml) was added *N*,*N*-diiisopropylethylamine (5.94 g, 46.05 mmol). After refluxing for 16 hr, the reaction mixture was cooled to room temperature, concentrated, and poured into water (20 ml). The resulting solid was filtered to obtain 500 mg of a mixture of *N*-(4-chloro-9*H*-pyrimido[4,5-*b*] indol-8-yl)formamide and 4-chloro-9*H*-pyrimido[4,5-*b*]indol-8-amine as a black solid. This mixture was taken in 4 N HCl in dioxane (15 ml). After stirring at room temperature for 4 hr, the reaction mixture was concentrated under reduced pressure, and the residue was adjusted to pH 7 with aq.$Na_2CO_3$, and extracted with EA (3 × 30 ml). The organic layers were dried over sodium sulfate, concentrated

under reduced pressure, and the residue was purified by reverse phase chromatography (water/acetonitrile/0.1% ammonium bicarbonate) to obtain 320 mg (10%) of 4-chloro-9H-pyrimido[4,5-b]indol-8-amine as a white solid. [1]H NMR (500 MHz, DMSO) δ 12.42 (s, 1H), 8.74 (s, 1H), 7.58 (d, $J$ = 7.8 Hz, 1H), 7.25–7.08 (m, 1H), 6.93 (d, $J$ = 7.7 Hz, 1H), 5.76 (s, 2H). LC–MS (ESI): $m/z$ = 219.2 (M+H)$^+$.

## AVI-4197/RLA-5830

To a solution of N-(4-chloro-9H-pyrimido[4,5-b]indol-8-yl)formamide and 4-chloro-9H-pyrimido[4,5-b]indol-8-amine (110 mg, 0.447 mmol), (R)-valinol (69.01 mg, 0.67 mmol) in DMSO (2 ml) was added triethylamine (171.6 mg, 1.41 mmol). After stirring at 100°C for 16 hr, the reaction mixture was extracted with ethyl acetate (3 × 20 ml), washed with brine (20 ml). The organic layer was dried over $Na_2SO_4$. The organic extracts were concentrated, and the residue was purified by silica gel column chromatography (50% ethyl acetate/petroleum ether) to obtain (R)-N-(4-((1-hydroxy-3-methylbutan-2-yl)amino)-9H-pyrimido[4,5-b]indol-8-yl)formamide as a white solid (45 mg, yield: 15.2%). LC–MS (ESI): $m/z$ = 314.3 (M+H)$^+$; RT = 1.30 min.

A solution of (R)-N-(4-((1-hydroxy-3-methylbutan-2-yl)amino)-9H-pyrimido[4,5-b]indol-8-yl)formamide (40 mg, 0.13 mmol) in HCl-dioxane (15 ml) was stirred at room temperature for 4 hr. The mixture was adjusted to pH 7 with aq.$Na_2CO_3$, and extracted with ethyl acetate (3 × 30 ml). The organic layer was dried over $Na_2SO_4$, the organic was concentrated and the residue was purified by reverse phase chromatography (0.1% $NH_4HCO_3$ in water, 10–100% ACN) to obtain (R)-2-((8-amino-9H-pyrimido[4,5-b]indol-4-yl)amino)-3-methylbutan-1-ol (AVI-4197) as a white solid (28.1 mg, yield: 70.52%). [1]H NMR (500 MHz, MeOD) δ 8.27 (s, 1H), 7.38 (d, $J$ = 7.8 Hz, 1H), 7.12 (t, $J$ = 7.8 Hz, 1H), 6.82 (d, $J$ = 7.7 Hz, 1H), 4.30–4.26 (m, 1H), 3.88 (dd, $J$ = 11.3, 4.8 Hz, 1H), 3.80 (dd, $J$ = 11.3, 4.0 Hz, 1H), 2.17 (d, $J$ = 7.1 Hz, 1H), 1.06 (dd, $J$ = 15.1, 6.8 Hz, 6H). LC–MS (ESI): $m/z$ 286.3 (M+H)$^+$.

## AVI-3367/RLA-5721

A mixture of 4-chloro-9H-pyrimido[4,5-b]indol-8-amine (28 mg, 0.13 mmol) and 1-aminopyrrolidin-2-one hydrochloride (35 mg, 0.26 mmol) in isopropanol/water (10:1, 1.1 ml) was heated to 100°C for 18 hr. The reaction mixture was filtered, the residue was washed with ethyl acetate and dried to obtain 28 mg (77%) of 1-((8-amino-9H-pyrimido[4,5-b]indol-4-yl)amino)pyrrolidin-2-one as brown solid. [1]H NMR (DMSO-$d_6$, 400 MHz) δ 12.99 (br s, 1H), 8.62 (s, 1H), 7.92 (br d, 1H, $J$ = 7.5 Hz), 7.27 (t, 1H, $J$ = 7.9 Hz), 7.05 (br d, 1H, $J$ = 7.5 Hz), 3.70 (br t, 2H, $J$ = 6.9 Hz), 2.44–2.53 (m, 2H), 2.20 (br t, 2H, $J$ = 7.4 Hz). [13]C NMR (METHANOL-$d_4$, 100 MHz) δ 175.9, 155.9, 154.3, 153.2, 132.5, 125.7, 121.9, 119.4, 111.3, 111.1, 97.0, 48.6, 47.9, 28.5, 15.9. LC–MS (ESI): $m/z$ = 283 (M+H)$^+$.

To a solution of 1-((8-amino-9H-pyrimido[4,5-b]indol-4-yl)amino)pyrrolidin-2-one (15 mg, 0.053 mmol) and triethylamine (0.015 ml, 0.11 mmol) in THF (1 ml), was added ethyl chloroformate (0.005 ml, 0.056 mmol). After stirring at 65°C for 18 hr, the reaction mixture was purified by reverse phase chromatography (water/acetonitrile/0.1% formic acid) to obtain 2.7 mg (13%) of ethyl (4-((2-oxopyrrolidin-1-yl)amino)-9H-pyrimido[4,5-b]indol-8-yl)carbamate formic acid salt (AVI-3367) as tan solid. [1]H NMR (METHANOL-$d_4$, 400 MHz) δ 8.42 (s, 1H), 7.94 (d, 1H, $J$ = 7.8 Hz), 7.59 (br s, 1H), 7.28 (t, 1H, $J$ = 7.9 Hz), 4.1–4.26–4.30 (m, 2H), 3.84 (t, 2H, $J$ = 7.1 Hz), 2.60 (t, 2H, $J$ = 8.0 Hz), 2.30–2.33 (m, 2H), 1.36–1.39 (m, 3H). LC–MS (ESI): $m/z$ = 355 (M+H)$^+$.

## (R)-2-((6-Bromo-7H-pyrrolo[2,3-d]pyrimidin-4-yl)amino)-3-methylbutan-1-ol

To a solution of 6-bromo-4-chloro-7H-pyrrolo[2,3-d]pyrimidine (900 mg, 3.9 mmol) in dry DMSO (10 ml) was added (R)-2-amino-3-methylbutan-1-ol (602 mg, 5.8 mmol) and TEA (787 mg, 7.8 mmol), the mixture was stirred at 110°C for 16 hr. LC–MS analysis showed the complete consumption of compound 6-bromo-4-chloro-7H-pyrrolo[2,3-d]pyrimidine. The mixture was diluted with ethyl acetate (40.0 ml) and washed with water (5.0 ml) and brine (5.0 ml). The organic layer was dried over $Na_2SO_4$ and concentrated under reduced pressure. The residue was purified by prep-HPLC (0.1% $NH_4HCO_3$ in water, 10–100% ACN) to give (R)-2-((6-bromo-7H-pyrrolo[2,3-d]pyrimidin-4-yl)amino)-3-methylbutan-1-ol as a white solid (522 mg, yield: 45%). [1]H NMR (500 MHz, DMSO-$d_6$) δ 12.22 (s, 1H), 8.03 (s, 1H), 7.00 (d, 1H, $J$ = 8.8 Hz), 6.79 (s, 1H), 4.62 (t, 1H, $J$ = 5.2 Hz), 4.13 (s, 1H), 3.52 (dd, 2H, $J$ = 9.4, 4.0 Hz), 1.98 (dt, 1H, $J$ = 13.6, 6.8 Hz), 0.91 (dd, 6 H, $J$ = 8.6, 6.9 Hz). LC–MS (ESI): $m/z$ = 299.2 (M+H)$^+$.

## AVI-4099 (RLA-5789)

A mixture of (R)-2-((6-bromo-7H-pyrrolo[2,3-d]pyrimidin-4-yl)amino)-3-methylbutan-1-ol (10.0 mg, 33.4 µmol), 5-(4,4,5,5-tetramethyl-1,3,2-dioxaborolan-2-yl)-1H-pyrazole (13.0 mg, 66.9 µmol), Pd(dppf)Cl$_2$ (4.9 mg, 6.7 µmol) and CsOH (12.5 mg, 83.6 µmol) in 0.25 ml of mixed solvent ($^n$BuOH/H$_2$O = 4/1) was stirred at 130°C for 20 min with microwave. The residue was purified by prep-HPLC (water, 0–30% ACN with 0.1% formic acid) to give (R)-2-((6-(1H-pyrazol-5-yl)-7H-pyrrolo[2,3-d]pyrimidin-4-yl)amino)-3-methylbutan-1-ol, formic acid salt (AVI-4099) as a white solid (3.7 mg, yield: 39%). $^1$H NMR (400 MHz, MeOD) (mixture of rotamers was observed) δ 8.42 (brs, 1H), 8.12 (brs, 1H), 7.73 (d, 1H, $J$ = 2.3 Hz), 6.97 (s, 1H), 6.72 (d, 1H, $J$ = 2.3 Hz), 4.16–4.11 (m, 1H), 3.84–3.75 (m, 2H), 2.16–2.06 (m, 1H), 1.09–1.02 (m, 6H). LC–MS (ESI): $m/z$ = 287 (M+H)$^+$.

## AVI-4211 (RLA-5849)

A mixture of (R)-2-((6-bromo-7H-pyrrolo[2,3-d]pyrimidin-4-yl)amino)-3-methylbutan-1-ol (15.0 mg, 50.1 µmol), phenylboronic acid (12.2 mg, 100.0 µmol), Pd(dppf)Cl$_2$ (3.7 mg, 5.01 µmol), and Cs$_2$CO$_3$ (40.8 mg, 125 µmol) in 0.22 ml of mixed solvent (dioxane/H$_2$O = 10/1) was stirred at 110°C for 17 hr. The residue was purified by prep-HPLC (water, 0–70% ACN with 0.1% formic acid) to give (R)-3-methyl-2-((6-phenyl-7H-pyrrolo[2,3-d]pyrimidin-4-yl)amino)butan-1-ol, formic acid salt (AVI-4211) as a white solid (9.7 mg, yield: 57%). $^1$H NMR (400 MHz, MeOD) (mixture of rotamers was observed) δ 8.41 (brs, 1H), 8.11 (brs, 1H), 7.79 (brd, 1H, $J$ = 8.0 Hz), 7.45 (brdd, 2H, $J$ = 8.0, 7.5 Hz), 7.33 (brt, 1 H, $J$ = 7.5 Hz), 7.03 (brs, 1H), 4.16–4.12 (m, 1H), 3.85–3.75 (m, 2H), 2.15–2.08 (m, 1H), 1.08–1.04 (m, 6H). LC–MS (ESI): $m/z$ = 297 (M+H)$^+$.

## AVI-372/RLA-5628

To a solution of 4-chloro-5-iodopyrimidine (400 mg, 1.66 mmol) in acetonitrile (5 ml) was added 1-aminopyrrolidin-2-one hydrochloride (250 mg, 1.84 mmol) and potassium carbonate (460 mg, 3.33 mmol). The reaction mixture was stirred at 80°C for 1 hr. The mixture was added water (15.0 ml) and extracted with ethyl acetate (30 ml *3). The combined organics were washed with brine (10 ml). The organic layer was dried over sodium sulfate and concentrated under reduced pressure. The residue was purified by silica gel column chromatography (10:1 dichloromethane/methanol) to afford 384 mg (76%) of 1-((5-iodopyrimidin-4-yl)amino)pyrrolidin-2-one. LC–MS (ESI): $m/z$ = 305.

To a solution of 1-((5-iodopyrimidin-4-yl)amino)pyrrolidin-2-one (20 mg, 0.066 mmol) in 1,4-dioxane (1 ml) was added 2-fluoro-6-(tributylstannyl)pyridine (26 mg, 0.066 mmol), copper (I) iodide (1.3 mg, 0.0066 mmol), triethylamine (0.028 ml, 0.2 mmol) and Pd(PPh$_3$)$_4$(7.6 mg, 0.0066 mmol). After stirring at 110°C for 18 hr, the reaction mixture was filtered through a celite pad and purified by reverse phase chromatography (water/acetonitrile/0.1% formic acid) to obtain 8 mg (40%) of 1-((5-(6-fluoropyridin-2-yl)pyrimidin-4-yl)amino)pyrrolidin-2-one formic acid salt (AVI-372) as a pale yellow oil. $^1$H NMR (METHANOL-d$_4$, 400 MHz) δ 8.71 (br s, 1H), 8.65 (br s, 1H), 8.56 (br s, 1H), 8.34 (br s, 1H), 7.66 (t, 1H, $J$ = 5.6 Hz), 3.68 (t, 2H, $J$ = 7.2 Hz), 2.47 (br t, 2H, $J$ = 8.0 Hz), 2.16–2.20 (m, 2H). LC–MS (ESI): $m/z$ = 274 (M+H)$^+$.

## AVI-411/RLA-5549

A mixture of 4,6-dichloropyrimidine (100 mg, 0.671 mmol, 1.0 equiv), tert-Butyl 5-amino-1H-indazole-1-carboxylate (157 mg, 0.671 mmol, 1.0 equiv) and NEt$_3$ (196 µl, 1.41 mmol, 2.1 equiv) in i-PrOH (3 ml) was stirred in the microwave at 100°C for 20 min. The reaction mixture was cooled and evaporated under reduced pressure. The residue was diluted with saturated NaHCO$_3$ solution (20 ml) and extracted with EtOAc (3 × 20 ml). The combined organic extracts were washed with water (2 × 20 ml), brine (1 × 40 ml), dried (MgSO$_4$), filtered and purified by silica gel chromatography (0–5% MeOH/DCM) to obtain 30.8 mg (19%) of N-(6-chloropyrimidin-4-yl)-1H-indazol-5-amine as a light yellow solid.

A mixture of N-(6-chloropyrimidin-4-yl)-1H-indazol-5-amine (30 mg, 0.12 mmol, 1.0 equiv) and 1-aminopyrrolidin-2-one hydrochloride (17 mg, 0.12 mmol, 1.0 equiv) in i-PrOH (0.4 ml) was stirred in the microwave at 100°C for 20 min. The reaction mixture was cooled and evaporated under reduced pressure. The residue was diluted with saturated NaHCO$_3$ solution (20 ml) and extracted with EtOAc (3 × 20 ml). The combined organic extracts were washed with water (2 × 20 ml), brine (1 × 40 ml), dried (MgSO$_4$), filtered and purified by reverse phase chromatography (water/MeCN/0.1% formic acid) to obtain 8.1 mg (21%) of 1-((6-((1H-indazol-5-yl)amino)pyrimidin-4-yl)amino)pyrrolidin-2-one as

a colorless oil. [1]H NMR (METHANOL-d$_4$, 400 MHz) δ 8.14 (s, 1H), 8.03 (s, 1H), 7.84 (d, 1H, $J$ = 1.7 Hz), 7.56 (d, 1H, $J$ = 8.8 Hz), 7.39 (dd, 1H, $J$ = 1.8, 8.9 Hz), 5.84 (s, 1H), 3.63 (t, 2H, $J$ = 7.1 Hz), 2.43–2.48 (m, 2H), 2.15 (t, 2H, $J$ = 7.7 Hz). LC–MS (ESI): $m/z$ = 283 (M+H)$^+$.

## AVI-1495 (RLA-5688)

A mixture of 5-bromo-4-chloro-7$H$-pyrrolo[2,3-$d$]pyrimidine (15.0 mg, 64.5 μmol), 1-(aminomethyl) cyclopropan-1-ol (13.3 mg, 129.0 μmol) in 0.22 ml of mixed solvent IPA/H$_2$O (10:1) was stirred at 100°C for 16 hr. The residue was purified by prep-HPLC (water, 0–40% ACN) to give 1-(((5-bromo-7$H$-pyrrolo[2,3-$d$]pyrimidin-4-yl)amino)methyl)cyclopropan-1-ol (AVI-1495), as a brown solid (6.3 mg, yield: 34%). [1]H NMR (400 MHz, MeOD) δ 8.13 (s, 1H), 7.18 (s, 1H), 3.74 (s, 2H), 0.82–0.78 (m, 2H), 0.74–0.71 (m, 2H). LC–MS (ESI): $m/z$ = 310 (M+H)$^+$.

## AVI-3571 (RLA-5703)

A mixture of 4-chloro-5-methyl-7$H$-pyrrolo[2,3-$d$]pyrimidine (15.0 mg, 89.5 μmol), 1-(aminomethyl) cyclobutan-1-ol (18.1 mg, 179.0 μmol) in 0.22 ml of mixed solvent IPA/H$_2$O (10:1) was stirred at 100°C for 4 days. The residue was purified by prep-HPLC (water, 0–5% ACN with 0.1% formic acid) to give 1-(((5-methyl-7$H$-pyrrolo[2,3-$d$]pyrimidin-4-yl)amino)methyl)cyclobutan-1-ol (AVI-3571), formic acid salt as a white solid (7.7 mg, yield: 31%). [1]H NMR (400 MHz, MeOD) δ 8.08 (s, 1H), 6.87 (s, 1H), 3.74 (s, 2H), 2.46 (s, 3H), 2.19–2.07 (m, 4H), 1.83–1.75 (m, 1H), 1.70–1.63 (m, 1H). LC–MS (ESI): $m/z$ = 233 (M+H)$^+$.

## AVI-1507 (RLA-5699)

To a solution of 4-chloro-7$H$-pyrrolo[2,3-$d$]pyrimidine (70 mg, 0.45 mmol) in dry DMSO (5 ml) was added ($R$)-pyrrolidin-2-ylmethanol (51 mg, 0.50 mmol) and TEA (227 mg, 2.25 mmol), the mixture was stirred at 110°C for 16 hr. The mixture was diluted with ethyl acetate (50.0 ml) and washed with water (10.0 ml), brine (10.0 ml). The organic layer was dried over Na$_2$SO$_4$ and concentrated under reduced pressure. The residue was purified by prep-HPLC (0.1% NH$_4$HCO$_3$ in water, 5–45% ACN) to give ($R$)-(1-(7$H$-pyrrolo[2,3-$d$]pyrimidin-4-yl)pyrrolidin-2-yl)methanol (AVI-1507) as a white solid (35 mg, yield: 35%). [1]H NMR (500 MHz, MeOD) δ 8.07 (d, $J$ = 5.4 Hz, 1H), 7.08 (d, $J$ = 3.6 Hz, 1H), 6.66 (d, $J$ = 3.6 Hz, 1H), 4.66–4.44 (m, 1H), 3.93 (d, $J$ = 8.8 Hz, 1H), 3.87–3.71 (m, 2H), 3.63 (dd, $J$ = 10.9, 6.5 Hz, 1H), 2.21–1.99 (m, 4H). LC–MS (ESI): $m/z$ = 219.1 (M+H)$^+$.

## Acknowledgements

This work was supported by a National Institutes of Health NIHGM145238 and Chan Zuckerberg Initiative Essential Open Software grant to JSF and NIH NIAID Antiviral Drug Discovery (AViDD) grant U19AI171110 to JSF, SAW, and ARR. We thank Nigel Moriarty for help with ligand refinement and feedback on the manuscript, Matthew Smith for help with ligand strain calculations, Paul Emsley for clarifications about ligand strain calculations, Seth Harris, Colin Grambow, and Emel Adaligil for feedback on macrocycle fitting and strain.

## Additional information

### Competing interests

Adam R Renslo: is a co-founder of TheRas, Elgia Therapeutics, and Tatara Therapeutics, and receives sponsored research support from Merck, Sharp and Dohme. Henry van den Bedem: is an employee of Atomwise Inc, but the work in this publication does not overlap with his role there. James S Fraser: is a consultant to, shareholder of, and receives sponsored research support from Relay Therapeutics and a consultant to and shareholder of Vilya Therapeutics. The other authors declare that no competing interests exist.

## Funding

| Funder | Grant reference number | Author |
|---|---|---|
| National Institute of General Medical Sciences | NIHGM145238 | James S Fraser |
| National Institute of Allergy and Infectious Diseases | U19AI171110 | Adam R Renslo<br>James S Fraser<br>Stephanie A Wankowicz |

The funders had no role in study design, data collection, and interpretation, or the decision to submit the work for publication.

## Author contributions

Jessica Flowers, Data curation, Software, Formal analysis, Validation, Investigation, Visualization, Methodology, Writing – original draft, Writing – review and editing; Nathaniel Echols, Software, Methodology, Writing – review and editing; Galen J Correy, Priyadarshini Jaishankar, Takaya Togo, Data curation, Writing – review and editing; Adam R Renslo, Project administration, Writing – review and editing; Henry van den Bedem, Software, Writing – review and editing; James S Fraser, Conceptualization, Supervision, Funding acquisition, Writing – original draft, Project administration, Writing – review and editing; Stephanie A Wankowicz, Conceptualization, Software, Formal analysis, Validation, Investigation, Writing – original draft, Project administration, Writing – review and editing

## Author ORCIDs

Jessica Flowers http://orcid.org/0000-0002-3501-8804
Nathaniel Echols http://orcid.org/0009-0007-1597-9302
Galen J Correy http://orcid.org/0000-0001-5155-7325
Takaya Togo https://orcid.org/0000-0003-0243-0760
Adam R Renslo http://orcid.org/0000-0002-1240-2846
Henry van den Bedem http://orcid.org/0000-0003-2358-841X
James S Fraser https://orcid.org/0000-0002-5080-2859
Stephanie A Wankowicz https://orcid.org/0000-0002-4225-7459

Reviewer #1 (Public review): https://doi.org/10.7554/eLife.103797.3.sa1
Reviewer #3 (Public review): https://doi.org/10.7554/eLife.103797.3.sa2
Author response https://doi.org/10.7554/eLife.103797.3.sa3

## Additional files

### Supplementary files

MDAR checklist

Supplementary file 1. Test set PDBs and associated statistics.

Supplementary file 2. High strain PDBs and associated statistics.

Supplementary file 3. Macrocycle PDBs and associated statistics.

Supplementary file 4. Removed Macrocycle PDBs.

Supplementary file 5. Fragment PDBs and associated statistics.

Supplementary file 6. X-ray data collection settings and statistics.

Supplementary file 7. CryoEM PDBs.

### Data availability

All data and code are avaliable in the github: https://github.com/ExcitedStates/qfit-3.0 (copy archived at *Riley et al., 2025*).

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
