## [Editor Report · eLife Assessment]

The work presents a **valuable** extension of qFit-ligand, a computational method for modeling conformational heterogeneity of ligands in X-ray crystallography and cryo-EM density maps. The authors provide **solid** evidence of improved capabilities through careful validation against the previous version, particularly in expanding ligand sampling within conformational space. Such improvements suggest practical utility for challenging applications, including macrocyclic compound modeling and crystallographic drug fragment screening.

---

## [Referee Report · Reviewer #1 (Public review)]

Summary:

Flowers et al describe an improved version of qFit-ligand, an extension of qFit. qFit and qFit-ligand seek to model conformational heterogeneity of proteins and ligands, respectively, cryo-EM and X-ray (electron) density maps using multiconformer models-essentially extensions of the traditional alternate conformer approach in which substantial parts of the protein or ligand are kept in place. By contrast, ensemble approaches represent conformational heterogeneity through a superposition of independent molecular conformations.

The authors provide a clear and systematic description of the improvements made to the code, most notably the implementation of a different conformer generator algorithm centered around RDKit. This approach yields modest improvements in the strain of the proposed conformers (meaning that more physically reasonable conformations are generated than with the "old" qFit-ligand) and real space correlation of the model with the experimental electron density maps, indicating that the generated conformers also better explain the experimental data then before. In addition, the authors expand the scope of ligands that can be treated, most notably allowing for multi conformer modeling of macrocyclic compounds.

Strengths:

The manuscript is well written, provides a thorough analysis, and represents a needed improvement of our collective ability to model small-molecule binding to macromolecules based on cryo-EM and X-ray crystallography, and can therefore has a positive impact on both drug discovery and general biological research.

Weaknesses:

Weaknesses were addressed during review. Overall, the demonstrated performance gains are modest.

Specific comments:

(1) The accuracy of initial placement may be critical. At the same time, in my experience ambiguous cases are quite common, for example with flat ligands with a few substituents sticking out or with ligands with highly mobile tails. There remain some questions regarding sensitivity to initial ligand placement, which individual users should check for.

---

## [Referee Report · Reviewer #3 (Public review)]

Summary:

The manuscript by Flowers et al. aimed to enhance the accuracy of automated ligand model building by refining the qFit-ligand algorithm. Recognizing that ligands can exhibit conformational flexibility even when bound to receptors, the authors developed a bioinformatic pipeline to model alternate ligand conformations while improving fitting and more energetically favorable conformations.

Strengths:

The authors present a computational pipeline designed to automatically model and fit ligands into electron density maps, identifying potential alternative conformations within the structures.

Weaknesses:

Ligand modeling, particularly in cases of poorly defined electron density, remains a challenging task. The procedure presented in this manuscript exhibits limitations in low-resolution electron density maps (lower than 2.0 Å) and low-occupancy scenarios. Considering that the maps used to establish the operational bounds of qFit-ligand were synthetically generated, it's likely that the resolution cutoff will be even stricter when applied to real-world data.

---

## [Author Response]

The following is the authors’ response to the original reviews

**Public Reviews:**

**Reviewer #1 (Public review):**
Summary:Flowers et al describe an improved version of qFit-ligand, an extension of qFit. qFit and qFit-ligand seek to model conformational heterogeneity of proteins and ligands, respectively, cryo-EM and X-ray (electron) density maps using multi-conformer models - essentially extensions of the traditional alternate conformer approach in which substantial parts of the protein or ligand are kept in place. By contrast, ensemble approaches represent conformational heterogeneity through a superposition of independent molecular conformations.The authors provide a clear and systematic description of the improvements made to the code, most notably the implementation of a different conformer generator algorithm centered around RDKit. This approach yields modest improvements in the strain of the proposed conformers (meaning that more physically reasonable conformations are generated than with the "old" qFit-ligand) and real space correlation of the model with the experimental electron density maps, indicating that the generated conformers also better explain the experimental data than before. In addition, the authors expand the scope of ligands that can be treated, most notably allowing for multi-conformer modeling of macrocyclic compounds.Strengths:The manuscript is well written, provides a thorough analysis, and represents a needed improvement of our collective ability to model small-molecule binding to macromolecules based on cryo-EM and X-ray crystallography, and can therefore have a positive impact on both drug discovery and general biological research.Weaknesses:There are several points where the manuscript needs clarification in order to better understand the merits of the described work. Overall the demonstrated performance gains are modest (although the theoretical ceiling on gains in model fit and strain energy are not clear!).

We thank the reviewer for their thoughtful review. To address comments, we have added clarifying statements and discussion points around the extent of performance gains, our choice of benchmarking metrics, and the “standards” in the field for significance. We expanded our analysis to highlight how to use qFit ligand in “discovery” mode, which is aimed at supporting individual modeling efforts. As we now write in the discussion:

“It is advisable to employ qFit-ligand selectively, focusing on cases with a moderate correlation between your input model and the experimental data, strong visual density in the binding pocket, high map resolution, or when your single-conformer ligand model is strained.”

Additionally, we note in the discussion:

“qFit-ligand primarily serves as a “thought partner” for manual modeling. Modelers still must resolve many ambiguities, including initial ligand placement, to fully take advantage of qFit capabilities. In active modeling workflows or large scale analyses, the workflow would only accept the output of qFit-ligand when it improves model quality. In cases where qFit-ligand degrades map-to-model fit and/or strain, we can simply revert to the input model. In practice, users can easily remove poorly fitting conformations using molecular modeling software such as COOT, while keeping the well modeled conformations, which is an advantage of the multiconformer approach over ensemble refinement methods.”

**Reviewer #2 (Public review):**
Summary:The manuscript by Flowers et al. aimed to enhance the accuracy of automated ligand model building by refining the qFit-ligand algorithm. Recognizing that ligands can exhibit conformational flexibility even when bound to receptors, the authors developed a bioinformatic pipeline to model alternate ligand conformations while improving fitting and more energetically favorable conformations.Strengths:The authors present a computational pipeline designed to automatically model and fit ligands into electron density maps, identifying potential alternative conformations within the structures.Weaknesses:Ligand modeling, particularly in cases of poorly defined electron density, remains a challenging task. The procedure presented in this manuscript exhibits clear limitations in low-resolution electron density maps (resolution > 2.0 Å) and low-occupancy scenarios, significantly restricting its applicability. Considering that the maps used to establish the operational bounds of qFit-ligand were synthetically generated, it's likely that the resolution cutoff will be even stricter when applied to real-world data.

We thank Reviewer #2 for their comments on the role of conformational flexibility and how our tool addresses the complexity involved in modeling alternative conformations. We agree that there are limitations at low resolution, limiting the application of our algorithm. That is the case with all structural biology tools. Automatically finding alternative conformations of ligands in high-resolution structures is an enhancement to the toolbox of ligand fitting. Expanding the algorithm to work with fragment screening data is important in this realm, as almost all of this data fits in the high-resolution range where qFit-ligand works best.

The reported changes in real-space correlation coefficients (RSCC) are not substantial, especially considering a cutoff of 0.1. Furthermore, the significance of improvements in the strain metric remains unclear. A comprehensive analysis of the distribution of this metric across the Protein Data Bank (PDB) would provide valuable insights.

We agree that the changes are small, partially because the baseline (manually modeled ligands) is very high. To provide additional evidence, we added evaluations using EDIAm, which is a more sensitive metric. In Figure 2 (page 10), representing the development dataset, we see more improvements above 0.1. With this being said, it is unclear what constitutes a ‘substantial’ improvement for either of these metrics, especially considering alternative conformations may only change the coordinates of a subset of ligands, just slightly improving the fit to density.

We agree that looking across the PDB on strain would provide valuable insight. To explore this, we looked to see how qFit-ligand could improve the fitting of deposited ligands with high strain (see section: Evaluating qFit-ligand on a set of structures known to be highly strained, Page 15). While only a subset of these structures had alternative conformers placed (24.6%), we observed that in this subset, the ligands often improved the RSCC and strain. This figure also demonstrates that while RSCC may not change much numerically, the alternative conformers explain previously unexplained density with lower energy conformers than what is currently deposited.

To mitigate the risk of introducing bias by avoiding real strained ligand conformations, the authors should demonstrate the effectiveness of the new procedure by testing it on known examples of strained ligand-substrate complexes.

See above.

**Recommendations for the authors:**

**Reviewer #1 (Recommendations for the authors):**
A - Specific comments:(1) It appears necessary to provide qFit-ligand with an initial model with the ligand already placed. This is not clear from the start of the introduction on page 3. It appears that ligand position is only weakly adjusted fairly late in the process, in step F of Figure 1. It seems, therefore, that the accuracy of initial placement is rather critical (see the example discussed on page 21). At the same time, in my experience, ambiguous cases are quite common, for example with flat ligands with a few substituents sticking out or with ligands with highly mobile tails. It would be helpful for the authors to comment on the sensitivity to initial ligand placement, either in the discussion or, better yet, in the form of an analysis in which the starting model position is randomly perturbed.

In our revised version, we have modified the introduction to clarify the necessity of including an initial ligand model (page 4).

“The qFit-ligand algorithm takes as input a crystal or cryo-EM structure of an initial protein-ligand complex with a single conformer ligand in PDBx/mmCIF format, a density map or structure factors (encoded by a ccp4 formatted map or an MTZ), and a SMILES string for the ligand.”

We also describe our sampling algorithm more clearly (see: Biasing Conformer Generation, page 6). Steps A-E generate many conformations (using RDKit), which are then selected/fit into experimental density (using quadratic programming). To help with additional shifting issues in the input ligand, after the first selection, we do additional rotation/translation of the generated conformers that are kept. We then do another round of fitting to the density (quadratic programming followed by mixed integer quadratic programming).

Given this sampling, we have not elected to do an additional computational experiment to test the “radius of convergence” or dependence on initial conditions. However, we outline the fundamental procedure here so that someone can build on the work and test the idea:

- Create single conformer models as we currently do

- randomly perturb the coordinates of the ligand by 0.1-0.3Å

- refine to convergence, creating a series of “perturbed, modified true positives” for each dataset

- Run qFit ligand

- Evaluate the variability in the resulting multi-conformer models

(2) Top of page 6 ("Biasing Conformer Generation"): the authors say "as we only want to generate ligands that physically fit within the protein binding pocket, we bias conformation generation towards structures more likely to fit well within the receptor's binding site". Apart from the odd redundancy of this sentence, I am confused: at the stage that seems to be referred to here (A-C in Figure 1) is the fit to the electron density already taken into account, or does this only happen later (after step E)?

Thank you for pointing this out. We have edited the statement to clarify it:

“To guide the conformation generation from the Chem.rdDistGeom based on the ligand type and protein pocket, we developed a suite of specialized sampling functions to bias the conformational search towards structures more likely to fit well into the receptor’s binding site.”

We do not consider the electron density during conformer generation (only selection from the generated conformers). The sampling is additionally biased by the type of ligand and the size of the binding pocket.

(3) qFit-ligand appears to be quite slow. Are there prospects for speedup? Can the code take advantage of GPUs or multi-CPU environments?

We agree with this. We have made some algorithmic improvements, most notably removing duplicate conformers based on root mean squared distance. This, along with parallelization, decreased the average runtime from ~19 minutes to ~8 minutes (see additional details: qFit-ligand runtime, page 8). We do not currently take advantage of GPU specific code.

(4) Section: Detection of experimental true positive multi-conformer ligands:a) Why are carbohydrate ligands excluded? This seems like an important class of ligands that one would like qFit to be able to treat! Which brings me to a related question: can covalently attached groups (e.g., glycosylation sites!) be modeled using qFit-ligand, or is qFit-ligand restricted to non-covalently bound groups?

Currently, qFit-ligand does not support covalently bound ligands, but this is an area of interest we are hoping to expand into. In the revised version, we added the non-covalently attached carbohydrates back into the true positive dataset. In Figure 4 (page 14), we show that qFit-ligand is able to improve fit to the experimental density in around 80% of structures, while also often reducing torsion strain (see additional details: qFit-ligand applied to unbiased dataset of experimental true positives, page 14).

b) "as well as 758 cases where the ligand model's deposited alternate conformations (altlocs) were not bound in the same chain and residue number" - I do not understand what this means, or why it leads to the exclusion of so many structures. Likewise, a number of additional exclusions are described in Figure S3. Some more background on why these all happened would be helpful. Are you just left with the "easy" cases?

Sometimes modelers will list the multiple conformations of a bound ligand as a separate residue within the PDB file, rather than as a single multiconformer model. For example, rather than writing a multiconformer LIG bound at A, 201 with altlocs ‘A’ and ‘B’, a modeler might write this instead as LIG, A, 201 and LIG A, 301. We initially excluded these kinds of structures. However, we agree that this choice resulted in the removal of many potentially valid true positives. We have since updated our data processing pipeline to include these cases, and they are examined in the updated manuscript.

c) I do not follow the argument made at the end of this section (last two paragraphs on page 9): "when using a single average conformation to describe density from multiple conformations, the true low-energy states may be ignored". I get that, but the conformations in the "modified true positives" dataset derive directly from models in which two conformations were modeled, so this cannot be the explanation for why qFit-ligand models result in somewhat lower average strain. It would seem that the paper could be served by providing examples where single conformations were modeled in deposited structures, but qFit detects multiple conformations.

We agree with this comment that the strain obtained from the modified true positives is likely higher than the deposited models. However, the modified structure is refined with a single conformation, and therefore changed from the deposited “A” conformation. Thus, the reduced strain observed in our qFit-ligand models relative to the modified true positives is not unexpected.

To expand our dataset, we also looked at deposited structures with high strain, all of which were modeled as single conformers. Here, we saw a decrease in strain when alternative conformers were placed (see section: Evaluating qFit-ligand on a set of structures known to be highly strained, page 15). Further, we provide an example from the XGen macrocycle dataset where a ligand initially modeled as a single conformer exhibited relatively high strain. After qFit‐ligand modeled a second conformation, the overall strain was reduced (Figure 6C, page 19; Figure 6—figure supplement 1C, page 59).

(5) Section: qFit-ligand applied to an unbiased dataset of experimental true positives Bottom of page 14: The paragraph starting with "qFit-ligand shows particular strength in scenarios with strong evidence..." is enigmatic: there's no illustration (unless it directly relates to the findings in Figure 4, in which case this should be more explicit). Since this points out when the reader will and will not benefit from using qFit-ligand, it should be clear what the authors are talking about.

This claim considers all the evidence presented in the manuscript, not necessarily one particular aspect of it. We advise using qFit-ligand when there is a moderate correlation between the input model and the experimental data, strong visual density in the binding pocket, high map resolution, and/or when your single conformer ligand model is strained. We have made all of these points clearer in the updated manuscript.

B - Section: qFit-ligand can automatically detect and model multiple conformations of macrocycles:This is an exciting extension of qFit-ligand, but some aspects of the analysis strike me as worrisome. Of the initial dataset of 150 structures, fewer than half make it all the way through analysis. It's hard to believe that this is a fully representative subset. Why, for example, could 29 structures not be refined against the deposited structure factors? Why does strain calculation (in RDKit?) fail on 30 ligands? What about the other 18 cases--why did these fail (in PHENIX?).

We agree that this is a striking number of failures, however, we note that they are not specific shortcomings of qFit-ligand (in fact, most are because standard structural biology and/or cheminformatics software fail on many PDB depositions). Therefore, these failures reflect broader limitations in standard bioinformatics and refinement restraint files when handling macrocycles. The strain calculator we used was not built for macrocycles, and after consulting with many experts in the field, the consensus was that no method works well with macrocycles. We discuss these issues in additional detail in the discussion (page 27):

“Additionally, our algorithm’s placement within the larger refinement and ligand modeling ecosystem highlighted other areas that need improvement. We note that macrocycles, due to their complicated and interconnected degrees of freedom, suffer acutely from the refinement issues, as demonstrated by the failure of approximately one-third of datasets in our standard preparation or post-refinement pipelines due to ligand parameterization issues. Many of these stemmed from problematic ligand restraint files, highlighting the difficulty of encoding the geometric constraints of macrocycles using standard restraint libraries. Improved force-field or restraints for macrocycles are desperately needed to improve their modeling.”

C - Minor issues:(1) "Fragment-soaked event maps" - this is a semantically strange section title!

We have updated the section title in our revised manuscript. The new title is ‘qFit-ligand recovers heterogeneity in fragment-soaked event maps’.

(2) Too many digits! All over the manuscript, percentages are displayed with 0.01% precision, while these mostly refer to datasets with ~150 structures. Shifting just one structure from one category to another changes these percentages by nearly 1%.

We have updated the sig figs in our revised manuscript.

(3) The authors are keen to classify decreases in RSCC as significant only when these changes exceed 0.1, but do not apply the same standard for increases. For instance, in Figure 4B if we were to classify improvements as significant if ΔRSCC > 0.1, there would be fewer significant improvements than decreases in performance (although it is visually clear that for most datasets things get better. Similarly, in Figure 5A if we were to classify improvements as significant if ΔRSCC > 0.1, qFit-ligand would only yield significant improvements for two out of 73 cases-not a lot).

We agree with the reviewer that there needs to be more consistency in our analysis of improvements/deteriorations. However, we note that operationally, when the decreases in model quality are observed, the modeler would simply reject the new model in favor of the input model. We have added to the discussion:

“In active modeling workflows or large scale analyses, the workflow would only accept the output of qFit-ligand when it improves model quality. In cases where qFit-ligand degrades map-to-model fit and/or strain, we can simply revert to the input model. In practice, users can easily remove poorly fitting conformations using molecular modeling software such as COOT, while keeping the well modeled conformations, which is an advantage of the multiconformer approach over ensemble refinement methods.”

There is generally no consensus in the field as to what might indicate a ‘significant’ change in RSCC, and any threshold we choose would be arbitrary. We note that in our manuscript, we had previously characterized a decrease in RSCC to be ‘significant’ if it exceeded 0.1. However, as there is no real scientific justification for this cutoff, or any cutoff, we moved away from this framing in the revised manuscript. Therefore, we just classify if we improve RSCC. For example, on page 9:

“qFit-ligand modeled an alternative conformation in 72.5% (n=98) of structures. Compared with the modified true positive models, 83.7% (n=113) of qFit-ligand models have a better RSCC and 77.0% (n=104) structures saw an improvement in EDIAm, representing an improved fit to experimental data in the vast majority of structures.”

In addition, we have conducted additional experiments using more sensitive metrics (EDIAm) to further illustrate qFit-ligand’s performance.

(4) Small peptides are not discussed as a class of ligands, although these are quite common.

Canonical peptides can be modeled with standard qFit. Non-canonical peptides present failure modes similar to the macrocycles discussed above, with a mix of ATOM and HETATM records and the need for custom cif definitions and link records. For these reasons we have not included an analysis outside of the macrocycle section. We have noted this caveat in the discussion:

“We note that even linear non-canonical peptides present similar failure modes to macrocycles, with a mix of ATOM and HETATM records and the need for custom cif definitions and link records. For these reasons, we did not include analysis on small peptide ligands; however, canonical peptides can be modeled with standard qFit [8].”

(5) Top of page 10: "while refinement improves": what kind of refinement does this refer to?

This refers to refinement with Phenix. We have updated this language to reflect this (page 8). “We refer to these altered structures as our ‘modified true positives’, which we use as input to qFit-ligand, and subsequent refinement using Phenix.”

(6) Bottom of page 11: "they often did" -> "it often did"

We have made this change in the revised version.

(7) Top of page 14: RMSDs and B factors do have units.

We have added the units in our revision.

(8) Top of page 24. In the generation of a composite omit map, why are new Rfree flags being generated? Did I misunderstand that?

*r_free_flags.generate=True* only creates R-free flags if they are not present in the input file as is the case for many (especially older) PDB depositions.

(9) Bottom of page 27: how large is the mask? Presumably when alt confs of the ligand are possible, it would be helpful for the mask to cover those?

We agree that this mask should be updated. In our revision, we define the mask around the coordinates of the full qFit-ligand ensemble. The same mask is used to calculate the RSCC of the input (single conformer) model versus the qFit-ligand model.

(10) Middle of page 29: "These structure factors are then used to compute synthetic electron density maps." - It is not clear whether the following three sentences are an explanation of the details of that statement or rather things that are done afterwards.

We clarify this in the manuscript (page 36).

“These structure factors are then used to compute synthetic electron density maps. To each of these maps, we generate and add random Gaussian noise values scaled proportionally to the resolution. This scaling reflects the escalation of experimental noise as resolution deteriorates, a common occurrence in real-life crystallographic data.”

(11) Chemical synthesis: I am not qualified to assess this and am surprised to see some much detail here rather than in some other manuscript. Are the corresponding structures deposited anywhere?

All of the structures we discuss in this manuscript are deposited in the PDB and listed in Supplementary Table 5.

**Reviewer #2 (Recommendations for the authors):**
The data should consistently present the number of structures that exhibit improvements or deterioration in particular metrics, like RSCC and strain, using a cutoff that should be significant. For instance, stating that "85.93% (n=116) of structures having a better RSCC in the qFit-ligand models compared to the modified true positive models" without clarifying the magnitude of improvement (e.g., a marginal increase of 0.01 in RSCC) lacks meaningful context. The figures should clearly indicate the specific cutoff values used for each metric. The accompanying text should provide a detailed explanation for the selection of these cutoff values, justifying their significance in the context of the study.Currently, there is no established consensus within the field on what constitutes a 'significant' improvement in RSCC or strain values. As such, we chose not to impose an arbitrary cutoff and just look at which structures improve RSCC. We also removed all language stating significance, as there isn’t a good standard in the field to assess significance. This is especially important as only improvements would be considered in an active modeling project. In cases where qFit ligand degrades the RSCC (or strain) to a large extent, the modeler would simply revert to the input model.In the first section of Results: "First, for all ligands, we perform an unconstrained search function allowing the generated conformers to only be constrained from the bounds matrix (Figure 1A). This is particularly advantageous for small ligands that benefit from less restriction to fully explore their conformational space. We then perform a fixed terminal atoms search function (Figure 1B)." It is unclear whether a fixed terminal atom search was conducted for each conformer generated in the initial step to further explore the conformational space. This aspect should be clarified to provide a more comprehensive understanding of the methodology.

Each independent conformer generation function (A-E) is initialized with only the input ligand model and runs in parallel with the other functions. These functions do not build on each other, but rather perturb the input molecule independently of one another. In our updated manuscript, we have clarified the methodology (page 6).

“First, in all cases, we perform an unconstrained search function (Figure 1A), a fixed terminal atoms search function (Figure 1B), and a blob search function (Figure 1C).”

Phrase: "We randomly sampled 150 structures and, after manual inspection of the fit of alternative conformations, chose 135 crystal structures as a development set for improving qFit-ligand." The authors should explain why they filtered 10% of the structures.

To develop qFit-ligand, we wanted to use a very high-quality dataset. We needed to know with some degree of certainty that if qFit-ligand failed to produce an alternate conformation (or generated conformations low in RSCC or high in strain), the failure was due to an algorithmic limitation rather than poor-quality input data. Therefore, after selection based on numerical metrics, we manually examined each ligand in Coot to observe if we believed the alternative conformers fit well into the density.